# KANO: KOLMOGOROV–ARNOLD NEURAL OPERATOR

**Jin Lee**[1]*      **Ziming Liu**[2]      **Xinling Yu**[1]      **Yixuan Wang**[3]

**Haewon Jeong**[1]      **Murphy Yuezhen Niu**[1]      **Zheng Zhang**[1]†

[1]University of California, Santa Barbara
[2]Massachusetts Institute of Technology
[3]California Institute of Technology

## ABSTRACT

We introduce *Kolmogorov–Arnold Neural Operator* (KANO), a dual-domain neural operator jointly parameterized by both spectral and spatial bases with intrinsic symbolic interpretability. We theoretically demonstrate that KANO overcomes the pure-spectral bottleneck of Fourier Neural Operator (FNO): KANO remains expressive over a generic position-dependent dynamics (variable coefficient PDEs) for any physical input, whereas FNO stays practical only to spectrally sparse operators and strictly imposes fast-decaying input Fourier tail. We verify our claims empirically on position-dependent differential operators, for which KANO robustly generalizes but FNO fails to. In the quantum Hamiltonian learning benchmark, KANO reconstructs ground-truth Hamiltonians in closed-form symbolic representations accurate to the fourth decimal place in coefficients and attains $\approx 6 \times 10^{-6}$ state infidelity from projective measurement data, substantially outperforming that of the FNO trained with ideal full wave function data, $\approx 1.5 \times 10^{-2}$, by orders of magnitude.

## 1 INTRODUCTION

Contemporary science and engineering increasingly operate in regimes where the effective dimensionality and complexity of phenomena and data overwhelm human-designed calibrations and approximations. This motivates data-centric modeling of governing dynamics from observations (Karniadakis et al., 2021; Wang et al., 2023; Carleo et al., 2019). For a learned model to be constituted as a scientific law, it should first *generalize* universally over a well-defined domain, and also should be *interpretable* so that the learned representations can be extracted and reused for verification, testing, and downstream simulation. Mathematically, physical dynamics are generalized as operators as they are often formalized through PDEs (Courant et al., 1963; Evans, 2022). A large and practically important subclass consists of variable coefficient PDEs, in which at least one term has a coefficient that varies by its variables (Gilbarg et al., 1977); we define physical dynamics governed by such PDEs as *position-dependent dynamics*, when one of the variables that varies the coefficient is position. Examples include fluid flow in media with spatially varying viscosity or conductivity (Kundu et al., 2024), and the Schrödinger equation with a potential that is a function of position operators (Sakurai & Napolitano, 2020). Scientific AI such as operator networks (Kovachki et al., 2023; Lu et al., 2021) that efficiently approximate a generic position-dependent dynamics with tractable interpretability are therefore valuable, which we recognize the absence and aim to fill the gap in this work.

An operator network approximates an arbitrary mapping between infinite-dimensional function spaces by first encoding functions into finite latent vectors and then learning the latent-to-latent map that represents the target operator (Lanthaler et al., 2022). DeepONet of Lu et al. (2021; 2019) implements the most general dense operator network where two neural networks learn both encoding and latent mapping directly from data, based on the theoretical foundation laid by Chen & Chen (1995). Fourier Neural Operator (FNO) of Li et al. (2020), on the other hand, hard-codes the encoding as pseudo-spectral projection with its spectrally diagonal kernels. FNO is provably and empirically superior when its hard-coded sparsity is optimal (Li et al., 2020; Kovachki et al.,

---

*hojin@ucsb.edu
†zhengzhang@ece.ucsb.edu

2021), but this spectral sparsity becomes maladaptive for position-dependent or otherwise spectrally dense dynamics (Koshizuka et al., 2024; Qin et al., 2024). In such cases, the model size required for a target accuracy can grow super-exponentially (Kovachki et al., 2021), and although the universal approximation guarantee still holds, realistic size FNO may only converge on an in-sample mapping that fails outside the training distribution. Numerous variants of FNO attempted to break this spectral bottleneck. Some have broadened spectral coverage by exploiting factorized (Tran et al., 2021) or multi-scale (You et al., 2024) spectral kernels, and others have injected local spatial kernels alongside the original spectral ones (Wen et al., 2022; Liu-Schiaffini et al., 2024; Liu et al., 2025). Yet all prior works still privilege the spectral basis and cannot achieve optimal sparsity in the spatial basis.

In parallel, interpretability has recently pivoted around Kolmogorov–Arnold Network (KAN) (Liu et al., 2024b;a), whose edges are trainable univariate functions and thus amenable to human inspection. Several works demonstrate data-driven scientific modeling with KAN: Chiparova & Popov (2025); Gashi et al. (2025) use KAN for system identification, and Koenig et al. (2024) replace the MLPs in Neural ODEs (Chen et al., 2018) with KANs, each reporting symbolic recovery of benchmark equations and parameters. KANs have also been explored within operator networks: Abueidda et al. (2025) employed KANs instead of MLPs in DeepONet and Xiao et al. (2024); Wang et al. (2025) augmented FNO with KANs. Despite performance gains however, prior KAN-based operator networks have not reported symbolic recovery of the learned operator, leaving the avenue of an interpretable operator network largely unexplored.

To address these research gaps, we introduce the **Kolmogorov–Arnold Neural Operator (KANO)**, an interpretable operator network jointly parameterized in both spatial and spectral bases with KAN sub-networks embedded in a pseudo-differential operator framework (Hörmander, 2007; Kohn & Nirenberg, 1965). The key insight is to represent each component of the operator in the basis where it is sparse: differential terms spectrally, localized terms spatially, to achieve the most compact and tractable representation. Our work offers three main contributions to the scientific AI community.

- First, we demonstrate the pure-spectral bottleneck of FNO with an illustrative example and theoretically analyze why FNO cannot converge closely as desired to a generic position-dependent dynamics (variable coefficient PDEs) with a practical parameter complexity.

- Second, we propose a novel framework of KANO that is expressive over a generic position-dependent dynamics with intrinsic symbolic interpretability. We provide theoretical analysis on KANO's dual-domain (spatial and spectral) expressivity along with the empirical evidences of KANO robustly generalizing on unseen input subspace when FNO fails to.

- Finally we validate the performance of KANO on some synthetic operators and a quantum simulation benchmark. KANO successfully recovered the closed-form formula accurately to the fourth decimal place in coefficients. Compared to the FNO baseline, KANO used only $0.03\%$ of the model parameters, but achieved an order lower relative loss $\ell_2$ in our synthetic operator benchmarks, and a four-order lower state infidelity in the quantum Hamiltonian learning benchmark.

To the best of our knowledge, our work is the first to demonstrate and quantify the symbolic recovery via KAN in operator learning. We shift the paradigm from mere *universal approximation* in operator learning toward the *universal generalization* of an operator network. Different from DeepOKAN (Abueidda et al., 2025) which replaces MLPs with KANs in DeepONet, our work achieves generalization over disjoint out-of-distribution subspace via a novel architecture design.

## 2 BACKGROUND

### 2.1 OPERATOR LEARNING AND FOURIER NEURAL OPERATOR

Operator learning approximates mapping between infinite-dimensional function spaces, $\mathcal{G} : \mathcal{A} \to \mathcal{U}$,[1] from function pairs $\{(\mathbf{a}_i \in \mathcal{A}, \mathbf{u}_i = \mathcal{G}(\mathbf{a}_i) \in \mathcal{U})\}_{i=1}^{N}$[2] (Kovachki et al., 2023; 2024b). An operator network $\mathcal{G}_\theta$ first encodes input $\mathbf{a}_i$ via *encoder* $\mathcal{E}_m : \mathcal{A} \to \mathbb{C}^m$ into a latent vector, then learns the *latent map* $\mathbf{T}_\theta : \mathbb{C}^m \to \mathbb{C}^{m'}$ which the output is reconstructed to approximate the label $\mathbf{u}_i$ via *reconstructor* $\mathcal{R}_{m'} : \mathbb{C}^{m'} \to \mathcal{U}$: i.e. $\mathcal{G}_\theta = \mathcal{R}_{m'} \circ \mathbf{T}_\theta \circ \mathcal{E}_m$ (Lanthaler et al., 2022). For fixed $(\mathcal{E}_m, \mathcal{R}_{m'})$, we can

---

[1] $\mathcal{A}$ and $\mathcal{U}$ are Banach function spaces (e.g., Sobolev spaces) defined on a bounded domain $D \subset \mathbb{R}^d$.

[2] In practice, each function is sampled on a discretized grid in $D$ and stored as a vector.

define the projection $\mathbf{\Pi}$ of an operator $\mathcal{G}$ as

$$\mathbf{\Pi}(\mathcal{G}) = \mathcal{R}_{m'} \circ \widehat{\mathbf{T}} \circ \mathcal{E}_m \quad \text{where} \quad \widehat{\mathbf{T}} \in \underset{\mathbf{T}:\,\mathbb{C}^m \to \mathbb{C}^{m'}}{\arg\min} \left\| \mathcal{G} - \mathcal{R}_{m'} \circ \mathbf{T} \circ \mathcal{E}_m \right\|. \tag{1}$$

DeepONet (Lu et al., 2021; 2019) learns $\mathcal{E}_m$, $\mathcal{R}_{m'}$, and $\mathbf{T}_\theta$ all with two sub-networks. FNO (Li et al., 2020), on the other hand, *hard-codes* $\mathcal{E}_m$ to be the truncated Fourier transform and $\mathcal{R}_m$ to be its band-limited inverse.

**Fourier Neural Operator (FNO).** Let the domain $D \subset \mathbb{R}^d$ be periodic and write the Fourier transform $\mathcal{F}$ of function $\mathbf{a}(\mathbf{x})$ as $\hat{\mathbf{a}}(\boldsymbol{\xi})$:

$$[\mathcal{F}\mathbf{a}](\boldsymbol{\xi}) = \widehat{\mathbf{a}}(\boldsymbol{\xi}) = \int_D \mathbf{a}(\mathbf{x})\, e^{-i\,\boldsymbol{\omega}\mathbf{x}}\, d\mathbf{x}, \qquad \boldsymbol{\omega} = 2\pi\boldsymbol{\xi} \in \mathbb{Z}^d. \tag{2}$$

For fixed set of retained modes $\boldsymbol{\xi}_i \in \Xi = \{\boldsymbol{\xi}_1, \ldots, \boldsymbol{\xi}_m\} \subset \mathbb{Z}^d$, truncated Fourier transform $\mathcal{F}_m : \mathcal{A} \to \mathbb{C}^m$ and its band-limited inverse $\mathcal{F}_m^{-1} : \mathbb{C}^m \to \mathcal{U}$ can be defined as:

$$\mathcal{F}_m(\mathbf{a}) = [\widehat{\mathbf{a}}(\boldsymbol{\xi}_1), \ldots, \widehat{\mathbf{a}}(\boldsymbol{\xi}_m)], \qquad \mathcal{F}_m^{-1}(\mathcal{F}_m\mathbf{a})(\mathbf{x}) = \sum_{j=1}^{m} \widehat{\mathbf{a}}(\boldsymbol{\xi}_j)\, e^{2\pi i\,\boldsymbol{\xi}_j \cdot \mathbf{x}}, \tag{3}$$

with a slight abuse of notation. A single Fourier layer $\mathcal{L}_{\text{FNO}}$ of FNO is written as:

$$\mathcal{L}_{\text{FNO}}(\mathbf{a})(\mathbf{x}) = \sigma\Big( \mathcal{F}_m^{-1}\Big( \mathbf{R}_{\theta_1}(\boldsymbol{\xi}) \cdot \mathcal{F}_m(\mathbf{a})(\boldsymbol{\xi}) \Big)(\mathbf{x}) + \mathbf{W}_{\theta_2} \cdot \mathbf{a}(\mathbf{x}) \Big) \tag{4}$$

with learnable spectral block-diagonal multiplier $\mathbf{R}(\boldsymbol{\xi})$, parametrized linear transformation $\mathbf{W}$, and point-wise nonlinear activation $\sigma$. FNO is comprised of iterative $\mathcal{L}_{\text{FNO}}$ between lift-up ($\mathcal{P}$) and projection ($\mathcal{Q}$) networks:

$$\mathcal{G}_\theta^{\text{FNO}}(\mathbf{a}) = \mathcal{Q} \circ \mathcal{L}_{\text{FNO}}^{(\ell)} \circ \cdots \circ \mathcal{L}_{\text{FNO}}^{(1)} \circ \mathcal{P}(\mathbf{a}). \tag{5}$$

In the perspective of the operator network formulation (1) (Lanthaler et al., 2022), FNO hard-codes its encoder $\mathcal{E}_m$ as $\mathcal{F}_m$ and reconstructor $\mathcal{R}_m$ as $\mathcal{F}_m^{-1}$, then learns the latent map $\mathbf{T}$ by its iterative layers of parametrized linear kernels interleaved by non-linear activations (Kovachki et al., 2021).

## 2.2 KOLMOGOROV–ARNOLD NETWORK

KAN (Liu et al., 2024b;a) replaces fixed node activations of traditional MLP with simple sum operations and train the learnable univariate 1D functions $\phi$ on edges. With layer width $n_l \to n_{l+1}$ and input field $\mathbf{x}^{(l)} \to \mathbf{x}^{(l+1)}$, a KAN layer yields a function matrix $\mathbf{\Phi}^{(l)}$ at $l^{\text{th}}$ layer as

$$\mathbf{x}^{(l+1)} = \mathbf{\Phi}^{(l)} \mathbf{x}^{(l)3}, \qquad \mathbf{\Phi}^{(l)} = \big[\, \phi_{q,p}^{(l)}(\cdot)\, \big]_{q=1,\ldots,n_{l+1}}^{p=1,\ldots,n_l}, \qquad x_q^{(l+1)} = \sum_{i=1}^{n_l} \phi_{q,p}^{(l)}(x_p^{(l)}), \tag{6}$$

so each output channel is a sum of edgewise transforms of the inputs (Liu et al., 2024b;a). In the original KAN each edge function is a spline expansion

$$\phi_{q,p}^{(\ell)}(t) = c_{q,p,0}^{(\ell)}\, b(t) + \sum_{i=1}^{g} c_{q,p,i}^{(\ell)}\, B_i(t), \tag{7}$$

with learnable coefficients for a fixed base 1D function $b(\cdot)$ and 1D B–spline basis $\{B_i\}$. Because every $\phi_{q,p}$ is a 1D curve, KANs are directly inspectable and amenable to visualization followed by symbolic regression. On expressivity, Wang et al. (2024) theoretically prove that KANs match MLPs up to constant depth and width factors; empirically, with appropriate optimization recipes, KANs and MLPs exhibit comparable scaling on PDE and operator benchmarks (Shukla et al., 2024). Hence, swapping a latent MLP for a KAN preserves expressivity while enabling symbolic readout.

## 3 THEORETICAL ANALYSIS ON FNO'S PURE-SPECTRAL BOTTLENECK

This section first illustrates the pure-spectral bottleneck of FNO. Then we provide a theoretical analysis and prove that FNO suffers the curse of dimensionality for position-dependent dynamics. FNO is proven to have the universal approximation guarantee over any arbitrary non-linear Lipschitz operator (Kovachki et al., 2021; Lanthaler et al., 2025). This section does not disprove the universal approximation ability of FNO; it illustrates the limitation on the *generalization ability* of FNO stemming from its pure-spectral bottleneck on spectrally dense operators.

---

[3]Akin to matrix-vector multiplication but follows the third equation Eq. 6 instead of row-vector inner product.

### 3.1 THE PURE-SPECTRAL BOTTLENECK OF FNO

We consider the 1D quantum harmonic oscillator Hamiltonian as an example:

$$\mathcal{H}a(x) = -\partial_{xx}a(x) + x^2 \cdot a(x). \tag{8}$$

Multiplication and differentiation have a dual relationship under the Fourier transform:

$$\mathcal{F}[(-\partial_{xx}a)](\xi) = \xi^2 \cdot \hat{a}(\xi), \qquad \mathcal{F}[(x^2 \cdot a)](\xi) = -\partial_{\xi\xi}\hat{a}(\xi). \tag{9}$$

In spectral domain, the spatial differential $\partial_{xx}$ is a spectral multiplier $\xi^2$, whereas the spatial multiplier $x^2$ becomes a spectral differential $\partial_{\xi\xi}$. Consider a truncated polynomial basis $\{1, x, x^2, \ldots, x^{n-1}\}$ and a truncated Fourier basis $e_k(\theta) = e^{ik\theta}$, $k = 0, \ldots, n-1$, on a periodic domain. In the spatial (polynomial) basis, the map $a(x) \mapsto x^2 \cdot a(x)$ acts as a two-step up-shift sparse matrix

$$\mathbf{S}_n^{(2)} := \begin{bmatrix} 0 & 0 & 1 & 0 & \cdots & 0 \\ 0 & 0 & 0 & 1 & \cdots & 0 \\ \vdots & \vdots & \vdots & \ddots & \ddots & 1 \\ 0 & 0 & 0 & \cdots & 0 & 0 \\ 0 & 0 & 0 & \cdots & 0 & 0 \end{bmatrix}, \tag{10}$$

while in the spectral (Fourier) basis it is a dense Toeplitz matrix (Morrison, 1995)

$$\mathbf{T}_n[x^2] := \begin{bmatrix} c_0 & c_{-1} & c_{-2} & \cdots & c_{-n+1} \\ c_1 & c_0 & c_{-1} & \cdots & c_{-n+2} \\ c_2 & c_1 & c_0 & \cdots & c_{-n+3} \\ \vdots & \vdots & \vdots & \ddots & \vdots \\ c_{n-1} & c_{n-2} & c_{n-3} & \cdots & c_0 \end{bmatrix}, \qquad c_m = \frac{1}{2\pi}\int_0^{2\pi}\theta^2 e^{-im\theta}\,d\theta. \tag{11}$$

Thus each term in $\mathcal{H}$ is sparse in one basis and dense in the other (Morrison, 1995).

An FNO layer $\mathcal{L}_{\text{FNO}}$ (4) can easily parametrize $-\partial_{xx}$ by taking $\mathbf{R}(\xi) \approx \xi^2$. However, approximating the dense *off*-diagonals in $\mathbf{T}_n[x^2]$ to parametrize $x^2$ must rely on the non-linear activation $\sigma(\cdot)$ since $\mathbf{R}(\xi)$ and $\mathbf{W}$ are spectrally diagonal and hence incapable of mixing modes. Let $\mathbf{z}(\mathbf{u})$ denote the pre-activation for input $\mathbf{u}$, then the Jacobian of $\mathcal{L}_{\text{FNO}}$ at $\mathbf{u}$ gives the first-order approximation of the learned map and its Fourier transform reveals the spectral off-diagonals of itself as

$$\mathcal{F}[\mathbf{J}(\mathbf{u})](\boldsymbol{\xi}, \boldsymbol{\xi}') = \Big(\mathcal{F}[\sigma'(\mathbf{z}(\mathbf{u}, \cdot))]\,[\boldsymbol{\xi} - \boldsymbol{\xi}']\Big) \cdot \Big(\mathbf{W} + \mathbf{R}_\theta(\boldsymbol{\xi}')\Big). \tag{12}$$

Therefore, *all* off-diagonals arise from the spectrum of the $\mathbf{u}$-dependent gate $\sigma'(\mathbf{z}(\mathbf{u}, \cdot))$: FNO's nonlinearity *can* create off-diagonals, but they are tied to the input distribution of $\mathbf{u}$. This is the pure-spectral bottleneck of FNO: *spectral off-diagonals of a learned FNO are tied to the train subspace, hence FNO can converge only on the in-sample mapping that fails outside the train distribution*[4]. See Appendix D for further detailed discussion expanding to the arbitrary higher order contribution and deep layered FNO.

### 3.2 FNO SUFFERS CURSE OF DIMENSIONALITY ON POSITION-DEPENDENT DYNAMICS

As explained previously, *position operator*, $a(x) \mapsto x \cdot a(x)$, is a highly dense Toeplitz map in the spectral basis (Morrison, 1995). Based on the Remark 21 & 22 of Kovachki et al. (2021), we prove that any position-dependent dynamics induces super-exponential scaling in FNO size by the desired error bound: *FNO cannot converge closely as desired on a generic position-dependent dynamics with practical model size, hence can only overfit on the in-sample mapping.* We provide Lemma 1, that a single position operator already spreads the input spectra too much for FNO to stay practical, and Theorem 1, expanding Lemma 1 to an arbitrary composition of position operators.

Following from the operator network formulation (1), the error estimate of an operator network $\mathcal{G}_\theta$ approximating the ground-truth operator $\mathcal{G}$ in an operator norm is bounded as

$$\|\mathcal{G} - \mathcal{G}_\theta\| \leq \underbrace{\|\mathcal{G} - \mathcal{R}_{m'} \circ \widehat{\mathbf{T}} \circ \mathcal{E}_m\|}_{\text{projection error}:\epsilon_{\text{proj}}} + \underbrace{\|\mathcal{R}_{m'} \circ (\widehat{\mathbf{T}} - \mathbf{T}_\theta) \circ \mathcal{E}_m\|}_{\text{latent network error}:\epsilon_{\text{net}}}, \tag{13}$$

by the triangle inequality. Latent network error $\epsilon_{\text{net}}$ follows the well-established scaling law of conventional neural networks (Hornik et al., 1989; Cybenko, 1989). Therefore, whether an operator network is efficient in model and sample size to achieve the desired accuracy hinges on the scalability of the projection error $\epsilon_{\text{proj}}$ (Lanthaler et al., 2022; Kovachki et al., 2021; 2024a).

---

[4]This issue of out-of-distribution fragility from underspecification is well studied by D'Amour et al. (2022)

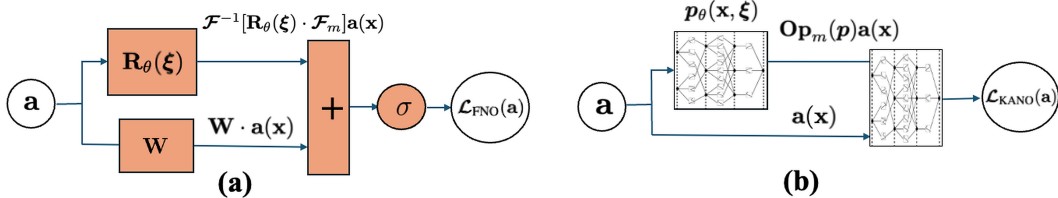

Figure 1: **(a)** $\mathcal{L}_{\text{FNO}}$ architecture. **(b)** $\mathcal{L}_{\text{KANO}}$ architecture.

**Reviewing Remark 21 & 22 of Kovachki et al. (2021)**  $\epsilon_{\text{proj}}$ of FNO is governed by the Fourier tail, the sum of Fourier coefficients outside the retained spectrum $\Xi$ truncated by width $m$ (Gottlieb & Orszag, 1977; Trefethen, 2000): to achieve the desired $\epsilon_{\text{proj}}$ with practical $m$, both input and output Fourier tails must decay algebraically or faster. However, even in the optimal case of the band-limited input, if the ground-truth operator is spectrally dense to spread out the input spectra and induce algebraic or slower decay in output Fourier tail, $m$ must scale at least polynomially to suppress $\epsilon_{\text{proj}}$: $m \sim \mathcal{O}\big(\epsilon_{\text{proj}}^{-1/s}\big)$ where $s$ is a geometric constant. Meanwhile, as the latent mapping would be also dense, the size of the latent network, $\mathcal{N}_{\text{net}}$, follows the canonical polynomial neural scaling (Yarotsky, 2017; De Ryck et al., 2021) by the desired $\epsilon_{\text{net}}$ with its width $m^d$ ($d$ is the input domain dimension) as the exponent: $\mathcal{N}_{\text{net}} \sim \mathcal{O}\big(\epsilon_{\text{net}}^{-m^d}\big)$. Consequently, this results as the super-exponential scaling in the latent network size $\mathcal{N}_{\text{net}} \sim \mathcal{O}\big(\epsilon_{\text{net}}^{-\epsilon_{\text{proj}}^{-d/s}}\big)$ even with the optimal band-limited input: *scaling width $m$ to suppress $\epsilon_{proj}$ explodes $\mathcal{N}_{net}$ to achieve the desired $\epsilon_{net}$ for a generic dense operator.*

**Lemma 1** (**Position operator elongates Fourier tail**). *A single position operator, spatial multiplier by $x$, induces algebraic decay in output Fourier tail when the input is band-limited.*

*Sketch of proof.* Position operator is kernel $\widehat{x}(\xi) \propto 1/\xi$ in spectral basis. Hence, every mode outside the input spectrum picks up a coefficient of size $\sim 1/|\xi|$, ending up as $|\hat{v}(\xi)| \gtrsim 1/|\xi|$ in the output spectrum. See Appendix C.2 for restatement and full proof. □

**Theorem 1** (**Curse of dimensionality on position operators**). *Any arbitrary composition of position operators requires FNO to scale super-exponentially on its model size by the desired accuracy.*

*Sketch of proof.* Iteratively apply Lemma 1, then any arbitrary composition of position operators induce algebraic or slower decay in output Fourier tail even for the optimal band-limited input. This results in super-exponential scaling of latent network size by the desired error as discussed above (Kovachki et al., 2021). See Appendix C.3 for restatement and full proof. □

**Remark 1.** *What is missing from the upper bound analysis of FNO by Kovachki et al. (2021) is the effect of the wide lift-up and projection networks. For the* generalization guarantee *arguments of this work, the upper bound analysis is still sound, yet it should be clarified that the theoretical analysis on the role of the lift-up and projection networks is yet an open research question. Recent studies by Lanthaler et al. (2025); Lanthaler (2024) provide better view on it with lower bound analysis.*

## 4 KOLMOGOROV−ARNOLD NEURAL OPERATOR

Motivated by the pure-spectral bottleneck of FNO, we propose the Kolmogorov-Arnold Neural Operator (KANO), an operator network capable of converging closely as desired on a generic position-dependent dynamics with practical model size. We first introduce the KANO architecture, and provide theoretical analysis on its dual-domain expressivity in the following section.

### 4.1 KANO ARCHITECTURE

KANO utilizes an iterative structure of KANO layers $\mathcal{L}_{\text{KANO}}$ to learn the unknown operator, akin to FNO. However, KANO excludes the wide lift-up and projection networks to maximize tractability since it is known that wide KANs are fragile to symbolic recovery (Noorizadegan et al., 2025):

$$\mathcal{G}_\theta^{\text{KANO}} = \mathcal{L}_{\text{KANO}}^{(\ell)} \circ \cdots \circ \mathcal{L}_{\text{KANO}}^{(1)}. \tag{14}$$

$$\mathcal{L}_{\text{KANO}}(\mathbf{a})(\mathbf{x}) = \Phi_{\theta_1}\Big(\mathcal{F}_m^{-1}\big[\, p_{\theta_2}(\mathbf{x}, \boldsymbol{\xi}) * \mathcal{F}_m(\mathbf{a})(\boldsymbol{\xi})\,\big](\mathbf{x})\,, \ \mathbf{a}(\mathbf{x})\Big), \tag{15}$$

where $\mathbf{\Phi}$ is a KAN sub-network for learnable non-linear activation; $\boldsymbol{p}(\mathbf{x}, \boldsymbol{\xi})$ is another KAN sub-network, a pseudo-differential symbol jointly parametrized by both spatial $\mathbf{x}$ and spectral $\boldsymbol{\xi}$ bases[5]. Note the "$*$" notation in $\mathcal{L}_{\text{KANO}}$ (15) instead of the block-diagonal multiplication notation "$\cdot$" in $\mathcal{L}_{\text{FNO}}$ (4). The spatial basis $\mathbf{x}$ of the symbol $\boldsymbol{p}(\mathbf{x}, \boldsymbol{\xi})$ is convolution (differential) in spectral domain by the dual relationship (9). Therefore, the pseudo-differential symbol calculus of $\boldsymbol{p}(\mathbf{x}, \boldsymbol{\xi})$ needs to be done by quantizing on both spatial and spectral domain (Hörmander, 2007), and we choose Kohn-Nirenberg quantization (Kohn & Nirenberg, 1965) to compute the symbol calculus in $\mathcal{L}_{\text{KANO}}$:

$$\mathcal{F}_m^{-1}\big[\,\boldsymbol{p}(\mathbf{x}, \boldsymbol{\xi}) \,*\, \mathcal{F}_m(\mathbf{a})(\boldsymbol{\xi})\,\big](\mathbf{x}) := \Big(\frac{h}{L}\Big)^d \sum_{\boldsymbol{\xi} \in \Xi} \sum_{\mathbf{y} \in \mathcal{Y}} e^{i(\mathbf{x}-\mathbf{y})\cdot\boldsymbol{\xi}}\, \boldsymbol{p}(\mathbf{x}, \boldsymbol{\xi})\, \mathbf{a}(\mathbf{y}), \qquad (16)$$

where for a periodic domain $D = (-\frac{L}{2}, \frac{L}{2})^d$, $\mathcal{Y} = \{\mathbf{y}_1, \ldots, \mathbf{y}_m\} \subset D$ is a uniform discretization with spacing $h$ and $\mathbf{x} \in D$ is an evaluation point. We denote Kohn-Nirenberg quantization (16) as an operator $\mathbf{Op}_m(\boldsymbol{p}) := \mathcal{F}_m^{-1}\big[\,\boldsymbol{p}(\mathbf{x}, \boldsymbol{\xi}) \,*\, \mathcal{F}_m\big]$ defined by the symbol $\boldsymbol{p}(\mathbf{x}, \boldsymbol{\xi})$. In the operator network formulation (1) introduced in Section 2.1, KANO's projection $\mathbf{\Pi}_{\text{KANO}}$ is then defined as:

$$\mathbf{\Pi}_{\text{KANO}}(\mathcal{G}) \;:=\; \mathbf{Op}_m(\boldsymbol{p}_{\mathcal{G}}), \qquad \boldsymbol{p}_{\mathcal{G}} \in \arg\min_{\boldsymbol{p}} \big\|\mathcal{G} - \mathbf{Op}_m(\boldsymbol{p})\big\|. \qquad (17)$$

**Symbolic Interpretability of KANO.** By using compact KANs each for the symbol $\boldsymbol{p}(\mathbf{x}, \boldsymbol{\xi})$ and non-linear activation $\mathbf{\Phi}$ in every KANO layer $\mathcal{L}_{\text{KANO}}$ (15), KANO network $\mathcal{G}_\theta^{\text{KANO}}$ (14) is fully inspectable by visualizing the learned edges of all its KANs, potentially allowing closed-form symbolic formula of the learned operator with the manual provided by Liu et al. (2024b;a). In addition, recent endeavors have greatly expanded KAN's symbolic recovery capacity to non-smooth, discontinuous targets with high irregularities (Yu et al., 2024; Aghaei, 2024; Lei et al., 2025; Yang et al., 2025; Shiraishi et al., 2025). All of such advancements are easily and directly applicable in our KANO framework as well, when facing an operator with high irregularity coefficients.

**Remark 2** (**Complexity analyses of KANO**). *As apparent in Eq. (16), KANO layer must perform double sum which can be computationally heavy. However, for the target operator class of variable-coefficient PDEs such as position-dependent dynamics, we show this can be compensated in principle by the parameter efficiency we prove in the following Section 4.2. See Appendix E.*

## 4.2  KANO'S DUAL-DOMAIN EXPRESSIVITY

In contrast to FNO, KANO exploits sparse representations in both the spatial and spectral domains, hence decoupling the scaling of $\epsilon_{\text{proj}}$ and $\epsilon_{\text{net}}$ by never letting the latent map be a dense convolution. For instance, for the quantum harmonic oscillator in Eq. (8), a KANO layer $\mathcal{L}_{\text{KANO}}$ (15) can parametrize $\mathcal{H}$ by taking $\boldsymbol{p}(x, \xi) \approx x^2 + \xi^2$, both $-\partial_{xx}$ and $x^2$ terms are each represented where they are sparse, both leveraging the shift form $\mathbf{S}_n^{(2)}$ (10). By *jointly* parameterizing the operator in both spatial and frequency domains, KANO *cherry-picks* the sparse representation for every term in position-dependent dynamics, building the right inductive bias well-known to be essential for out-of-distribution generalization and model efficiency (Goyal & Bengio, 2022; Trask et al., 2018).

This dual-domain expressivity of KANO first alleviates the input constraint; we first explain that $\epsilon_{\text{proj}}$ of KANO scales practically by its width $m$ for *any* physical input. Then we provide Theorem 2: as long as the KANO projection (17) of an operator generates smooth symbols KAN can easily approximate, $\epsilon_{\text{net}}$ scales practically by compact KAN sub-networks independent of $\epsilon_{\text{proj}}$. In conclusion, *KANO can converge closely as desired to a generic position-dependent dynamics with practical model size using any physical input, robustly generalizing outside the train subspace.*

**KANO practically has no input constraint.** According to the quadrature bound from Demanet & Ying (2011, Thm. 1&2), the error estimate of Kohn-Nirenberg quantization (16) obeys

$$\big\|\mathcal{G} - \mathbf{Op}_m(\boldsymbol{p}_{\mathcal{G}})\big\| \;\leq\; C\,B\,m^{-s}, \qquad (18)$$

given norm-bound (finite-energy)[6] input of $\mathcal{A}_B = \{\mathbf{u} \,:\, \|\mathbf{u}\| \,\leq\, B\}$ where $s, C$ are geometric constants. Hence KANO width $m$ scales polynomially by the desired $\epsilon_{\text{proj}}$ given any physical data.

---

[5]Shin et al. (2022) first employed pseudo-differential operator framework for neural operator. They presumed the symbol $\boldsymbol{p}(\mathbf{x}, \boldsymbol{\xi})$ to be separable as $\boldsymbol{p}(\mathbf{x}, \boldsymbol{\xi}) = \boldsymbol{p}_{\mathbf{x}}(\mathbf{x}) \cdot \boldsymbol{p}_{\boldsymbol{\xi}}(\boldsymbol{\xi})$, and used MLP sub-networks while retaining the lift-up and projection networks of a generic neural operator architecture (Kovachki et al., 2023).

[6]Norm here and Equation Eq. 18 is the Sobolev norm

Table 1: Relative $\ell_2$ losses ($\times 10^{-4}$) and parameter counts.

| | $\mathcal{G}_1$ | | | $\mathcal{G}_2$ | | | $\mathcal{G}_3$ | | |
|---|---|---|---|---|---|---|---|---|---|
| Model (params) | A | B | B/A | A | B | B/A | A | B | B/A |
| FNO (566k) | 6.36 | 98.8 | 15.53 | 10.6 | 87.0 | 8.21 | 11.4 | 81.4 | 7.14 |
| U-FNO (579k) | 2.79 | 22.9 | 8.21 | 8.14 | 339 | 41.65 | 92.4 | 292 | 3.16 |
| AM-FNO (548k) | 1.08 | 20.9 | 19.35 | 1.20 | 16.5 | 13.75 | 1.16 | 29.8 | 25.69 |
| PDNO (538k) | 1.41 | 6.31 | 4.5 | 1.92 | 12.1 | 6.3 | 4.03 | 27.2 | 6.7 |
| KANO (152) | 1.04 | 1.44 | 1.38 | 0.629 | 0.749 | 1.19 | 0.716 | 0.737 | 1.03 |
| KANO$_{\text{MLP}}$ (2k) | 3.37 | 6.59 | 1.96 | 4.49 | 8.07 | 1.80 | 3.59 | 6.87 | 1.91 |
| KANO$_{\text{SYMBOLIC}}$ | 0.512 | 0.526 | 1.03 | 0.498 | 0.500 | 1.00 | 0.520 | 0.536 | 1.03 |

Figure 2: Loss test results. **(a)** $\mathcal{G}_1$ **(b)** $\mathcal{G}_2$ **(c)** $\mathcal{G}_3$. Note the logarithmic scale.

**Theorem 2** (**KANO stays practical for smooth symbol**). *If the KANO projection of an operator* $\mathcal{G}$, $\Pi_{KANO}(\mathcal{G})$ *(17), generates a finite composition of smooth symbols* $p_{\mathcal{G}}(\mathbf{x}, \boldsymbol{\xi})$ *and finite-degree non-linearities, the model size of KANO scales polynomially by the desired accuracy* $\varepsilon$.

*Sketch of proof.* Choosing $m \sim (B/\varepsilon)^{1/s}$ scales projection error down to $\varepsilon/2$ by Eq. (18). A fixed-width KAN then approximates the symbols to accuracy $\varepsilon/2$ with $\mathcal{O}(\varepsilon^{-d/(2s_p)})$ parameters (Wang et al., 2024, Corol. 3.4) ($s_p$ is a geometric constant). The finite-degree non-linearities add only constant-size weights by the activation KAN, so the total parameter count is $\mathcal{O}(\varepsilon^{-d/(2s_p)})$. See Appendix C.4 for restatement and full proof. □

**Corollary 1** (**KANO is practical for generic position-dependent dynamics**). *For a finite composition of spatial and spectral multipliers of maximum* $r$-differentiable symbols with finite-degree non-linearity, Theorem 2 yields $|\Theta| = \mathcal{O}(\varepsilon^{-d/(2r)})$.

**Remark 3** (**Scope of KANO**). *Recent studies demonstrate that wide lift-up and projection networks are essential for strong performance on high-dimensional benchmarks (Diab & Al Kobaisi, 2025; Eker, 2024; Liu et al., 2023). In contrast, KANO is designed to prioritize symbolic recovery with robust generalization, complementary to the scope of FNO. Because the core dual-domain expressivity is mathematically agnostic to dimensionality, extending KANO to conventional high-dimensional use cases is a natural and promising direction for future work.*

## 5 EXPERIMENTAL RESULTS

### 5.1 SYNTHETIC-OPERATOR GENERALIZATION BENCHMARKS

We benchmark FNO-based models and KANO on three position-dependent operators:

$$\mathcal{G}_1 f = x^2 \cdot f \; - \; \partial_{xx} f, \quad \mathcal{G}_2 f = x \cdot \partial_x f \; + \; \partial_{xx} f, \quad \mathcal{G}_3 f = f^3 + x \cdot \partial_x f + \partial_{xx} f.$$

Our goal is to quantify and compare the generalization of each model. We train the models *only* with Group A dataset and evaluate them on the *unseen* Group B dataset.

- **Group A (Training families):** Periodic, Chirped Cosine, Sine Beats.
- **Group B (Testing families):** Sinc Pulse, Gaussian×Hermite, Wave Packet.

For each operator, we generate 2000 train pairs from Group A and 400 test pairs from Group B to evaluate the generalization by comparing the ratio between the average relative $\ell_2$ loss over each group (**Loss Test**). We also interpolate the Group A and B function samples in 100 steps, apply ground-truth operators in each step to build the interpolated dataset, and evaluate the loss ratio to that of the Group A samples (**Interpolation Test**). We trained FNO models of 2 layers, 64 width with no

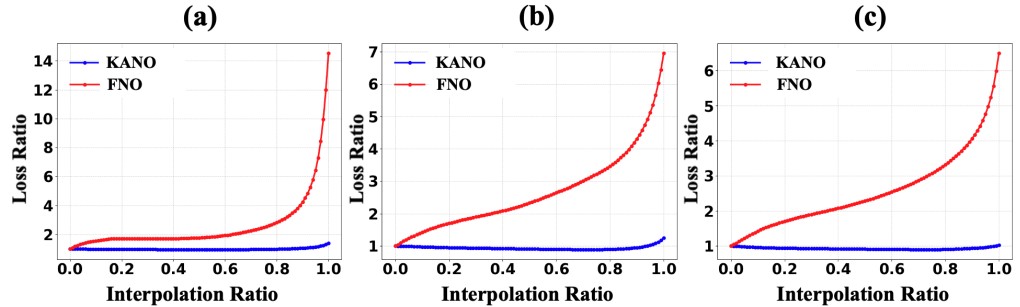

Figure 3: Interpolation test results. **(a)** $\mathcal{G}_1$ **(b)** $\mathcal{G}_2$ **(c)** $\mathcal{G}_3$.

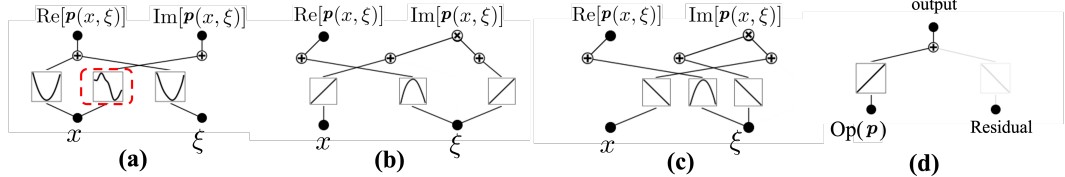

Figure 4: **(a)** $p(x,\xi)$ of $\mathcal{G}_1$. The middle edge does not contribute to the output. **(b)** $p(x,\xi)$ of $\mathcal{G}_2$. **(c)** $p(x,\xi)$ of $\mathcal{G}_3$. **(d)** $\Phi$ of $\mathcal{G}_3$. Edge of the residual in **(d)** looks linear, so we compared two scenarios, linear and cubic, which the latter achieved lower loss and better generalization.

mode truncation, and used one-layer KANs of grid 10 cubic B-splines edges for the KANO model. For other FNO variant baselines, we used 2-layer models of similar size to FNO. Lastly, we trained a KANO variant for an ablation study, which we replaced the KAN subnetworks with compact MLPs of 32 hidden width and 2 hidden layers (KANO_MLP). See Appendix B.1 for experiment details. We used Adam optimizer and relative $\ell_2$ loss for training.

**Results.** As shown in Table 1 and Figure 2, KANO shows consistent losses over Group A and Group B, validating its robust generalization ability for position-dependent operators. KANO_MLP also shows comparable out-of-distribution performance, which suggests that the generalization ability of KANO stems from its dual-domain architecture apart from KAN. In contrast, FNO shows fragile out-of-distribution behavior on Group B dataset with the significant loss increases. U-FNO (Wen et al., 2022) and AM-FNO (Xiao et al., 2024) show even worse results. On the other hand, PDNO (Shin et al., 2022) shows the most stable generalization among FNO families, although not as robust as KANO and KANO_MLP. Along with the ablation study, this confirms that the pseudo-differential operator framework is judicious for robust generalization on position-dependent operators, while primarily relying on only spectral kernels makes the model fragile out of train distribution even with localized enhancements. Interpolation test results in Figure 3 further empirically validate our theories. The FNO curves (red) of Figure 3 show slow increases on early and mid-interpolation, suggesting that the FNO's learned in-sample mappings are yet close to the ground-truth operators. However, the FNO curves abruptly soar up in the latter ratio, suggesting that the interpolated functions are now far outside the train distribution. These results, together with KANO's one-order–of–magnitude lower loss at just $0.03\%$ of FNO's size, are consistent with our claims in Theorem 1 and Theorem 2.

After convergence, we visualized the embedded KANs (Figure 4). We then froze these learned symbols and continued training, referring to this variant as KANO_symbolic. KANO_symbolic recovered the exact symbolic coefficients of the ground-truth operator to within the fourth decimal place (Table 2). KANO's loss matches KANO_symbolic's loss in Table 1, confirming that KANO converged close to the ground-truth operator.

## 5.2 LONG-HORIZON QUANTUM DYNAMICS BENCHMARK

We provide this benchmark on two position-dependent quantum dynamics: the quartic double-well Hamiltonian (DW) and the nonlinear Schrödinger equation with cubic nonlinearity (NLSE):

$$i\partial_t \psi = -\tfrac{1}{2}\partial_{xx}\psi + w(x)\cdot\psi \ \text{(DW)}, \quad i\partial_t \psi = -\tfrac{1}{2}\partial_{xx}\psi + w(x)\cdot\psi + |\psi|^2\cdot\psi \ \text{(NLSE)},$$

Table 2: Ground-truth vs. learned operators (coefficients rounded to $4^{th}$ decimal place).

| Ground-truth operator | Learned KANO operator |
|---|---|
| $\mathcal{G}_1 f = x^2 \cdot f - \partial_{xx} f$ | $\tilde{\mathcal{G}}_1 f = (x^2 + 0.0003) \cdot f - \partial_{xx} f$ |
| $\mathcal{G}_2 f = x \cdot \partial_x f + \partial_{xx} f$ | $\tilde{\mathcal{G}}_2 f = 0.9996\, x \cdot \partial_x f + \partial_{xx} f - 0.0003\, f$ |
| $\mathcal{G}_3 f = f^3 + x \cdot \partial_x f + \partial_{xx} f$ | $\tilde{\mathcal{G}}_3 f = 1.0001\, f^3 + 0.99997\, x \cdot \partial_x f + 0.99997\, \partial_{xx} f$ |
| | $\qquad - 0.0002\, f^2 - 0.0003\, f - 0.0001$ |

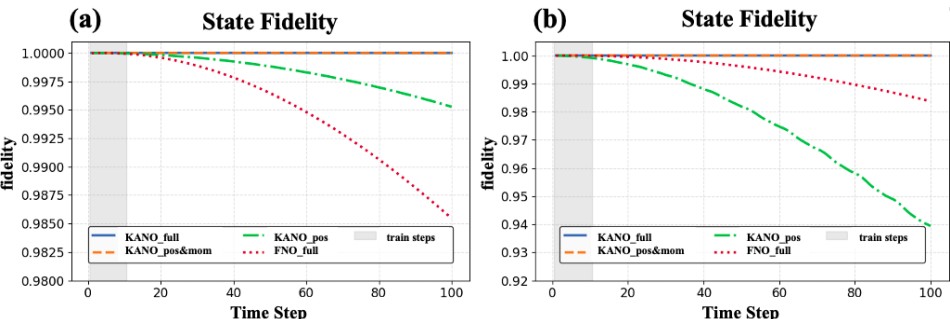

Figure 5: State fidelity over 100 time steps. **(a)** DW **(b)** NLSE

where $w(x) = x^4 - \left(x - \frac{1}{32}\right)^2 + 0.295$.

We generate 200 initial states and yield the state trajectories by the Hamiltonians, sampling momentum/position probability mass functions (PMFs) every 0.1ms for 100 time steps. The first 10 time steps are used for training, and the rest are used to evaluate the long-horizon prediction.

We modify KANO to capture the quantum state evolution: **Q-KANO**. Symbol $p_\theta$ is parametrized as $\exp\left[-i\Delta T\, \phi_\theta(\mathbf{x}, \boldsymbol{\xi})\right]$, where $\Delta T = 0.1$ ms. The adaptive activation is also defined as a complex exponential with learned phase $\boldsymbol{\vartheta} = \boldsymbol{\Phi}_\theta(|\mathbf{Op}_m(p_\theta)\psi|, \angle\mathbf{Op}_m(p_\theta)\psi)$ for input wave function $\psi(\mathbf{x})$:

$$\mathcal{G}_\theta^{\text{Q-KANO}}[\psi] = \mathbf{Op}_m(\exp\left[-i\Delta T\, \phi_\theta(\mathbf{x}, \boldsymbol{\xi})\right])\psi \,\cdot\, e^{-i\Delta T\, \boldsymbol{\vartheta}}. \qquad (19)$$

We investigate three supervision scenarios: **Full**-type training with full wave function, idealistic yet physically unattainable, **Pos**-type training with only position PMF, physically realistic yet the least informative, and **pos&mom**-type training with both position *and* momentum PMFs, which remains physically attainable while providing richer information although not full. We use Adam optimizer for all trainings. See Appendix B.2 for experiment details.

**Results.** We evaluate state infidelity[7] between ground-truth evolution and model prediction at each time step (Table 3, Figure 5). In case of KANO, the **pos & mom**-type training achieves indistinguishable infidelity from the ideal **full**-type training baseline. The ablation study with MLP variant of Q-KANO achieved comparably low state

Table 3: State infidelity after 90 additional time-evolution steps.

| Model & Train Type | State Infidelity | |
|---|---|---|
| | Double-Well | NLSE |
| FNO (full) | $1.5 \times 10^{-2}$ | $1.6 \times 10^{-2}$ |
| Q-KANO (full) | $6.3 \times 10^{-6}$ | $6.8 \times 10^{-6}$ |
| Q-KANO (pos & mom) | $6.3 \times 10^{-6}$ | $5.6 \times 10^{-6}$ |
| Q-KANO (pos) | $4.7 \times 10^{-3}$ | $6.1 \times 10^{-2}$ |
| Q-KANO_MLP (pos & mom) | $7.7 \times 10^{-6}$ | $8.5 \times 10^{-6}$ |
| Q-KANO_SYMBOLIC (full) | $2.0 \times 10^{-8}$ | $2.0 \times 10^{-8}$ |
| Q-KANO_SYMBOLIC (pos & mom) | $2.0 \times 10^{-8}$ | $3.0 \times 10^{-8}$ |
| Q-KANO_SYMBOLIC (pos) | $5.3 \times 10^{-2}$ | $6.1 \times 10^{-2}$ |

infidelity by **pos & mom**-type training as well. Meanwhile, the **pos-type** training displays a clear increase in infidelity, especially on the NLSE.

---

[7]For predicted state $\tilde{\boldsymbol{\varphi}}$ and ground-truth state $\boldsymbol{\varphi}$, the state fidelity $F$ is defined as the inner product between them ($F := <\tilde{\boldsymbol{\varphi}}, \boldsymbol{\varphi}>$), and the state infidelity is defined as $(1 - F)$, hence shows how distant two states are.

Table 4: Ground truth vs. learned symbols. Coefficients rounded to $4^{th}$ decimal place.

| Hamiltonian | Train Type | Learned symbolic structure |
|---|---|---|
| **DW** | ground truth | $x^4 - x^2 + 0.0625\,x + 0.295 + 0.5\,\xi^2$ |
| | full | $1.0004\,x^4 + 0.0001\,x^3 - 1.0013\,x^2 + 0.0625\,x + 0.2955 + 0.5\,\xi^2$ |
| | pos & mom | $1.0003\,x^4 + 0.0001\,x^3 - 1.0008\,x^2 + 0.0623\,x + 0.0001 + 0.5\,\xi^2$ |
| **NLSE** | ground truth | $x^4 - x^2 + 0.0625\,x + 0.295 + 0.5\,\xi^2 + |\psi|^2$ |
| | full | $1.0005\,x^4 - 0.0001\,x^3 - 1.0014\,x^2 + 0.0626\,x + 0.2942 + 0.5\,\xi^2 + 0.9815|\psi|^2 + 0.0110|\psi|$ |
| | pos & mom | $0.9999\,x^4 - 0.0003\,x^3 - 1.0001\,x^2 + 0.0630\,x + 0.1141 + 0.5\,\xi^2 + 0.9514|\psi|^2 - 0.5504|\psi|$ |

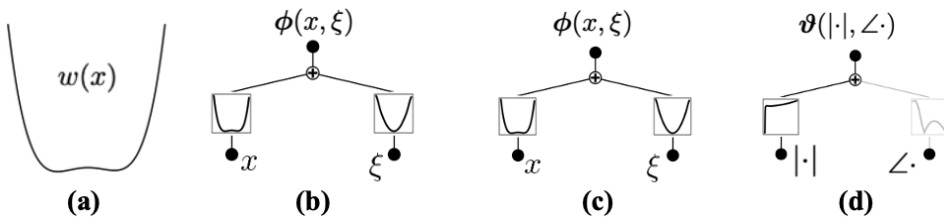

Figure 6: **pos&mom** type training results. **(a)** Structure of the potential $w(x)$ **(b)** $p(x,\xi)$ of DW. **(c)** $p(x,\xi)$ of NLSE. **(d)** $\varphi(|\cdot|, \angle\cdot)$ of NLSE. Potential $w(x)$ structure is clearly reconstructed.

In contrast, even with **full** type training, FNO fails to maintain low state infidelity after the long-horizon propagation as expected. Iterative time evolution pushes the wave function far outside the train convex hull, and FNO's learned in-sample mapping deviates from the ground-truth evolution rapidly, leading to four orders of infidelity increase compared to KANO.

Table 4 juxtaposes the learned symbols with that of the ground-truth Hamiltonians and Figure 6 shows the KAN visualizations from **pos & mom**-type training. With **full**-type training coefficients are recovered to the fourth decimal place, vindicating the ideal capacity of KANO when the information is fully provided. Under the realistic **pos & mom**-type training, the reconstruction remains accurate except for two terms: the constant (global phase) and the NLSE's cubic coefficient. Both discrepancies are predicted by quantum observability: global phases cancel in all PMFs, and the Kerr coefficient enters only through higher-order correlations that become harder to estimate from finite-shot statistics. Q-KANO faithfully reveals what the data support and nothing more.

## 6 CONCLUSION

We have presented the Kolmogorov–Arnold Neural Operator, an interpretable neural operator expressive on a generic position-dependent dynamics. KANO cherry-picks sparse representations of each term via jointly parametrizing on both spectral and spatial bases, and achieves robust generalization outside train distribution while exposing clear tractable representation via its KAN sub-networks. In all our benchmarks, KANO has successfully recovered the ground-truth operators accurately to the fourth decimal place in coefficients. In addition to the superior out-of-distribution generalization, KANO has also achieved orders of magnitude lower losses with less than $0.03\%$ of the model size compared to the FNO baseline. KANO shifts operator learning from an opaque, surrogate-based paradigm towards interpretable data-driven scientific modeling, and provides robust empirical evidence supporting its enhanced dual-domain expressivity and interpretability.

## REPRODUCIBILITY STATEMENT

All code, configurations, and dataset artifacts required to reproduce the experiments are provided in https://github.com/jinucsb/KANO, along with step-by-step instructions for rerunning training and evaluation. Details in the data generation and implementation are provided in Appendix B,

and the full proofs of Theorems and Lemma are provided in Appendix C along with their mathematical restatements.

## ETHICS STATEMENT

All authors of this work sincerely adhered to the ICLR Code of Ethics. We do not expect any potential violation to the best of our knowledge.

## ACKNOWLEDGMENTS

We thank Nina Miolane, Adele Myers, Louisa Cornelis for helpful early discussions and Anima Anandkumar for her feedback on our paper draft. Jin Lee and Zheng Zhang were supported by NSF ECCS-2328281. Murphy Yuezhen Niu was supported by the U.S. National Science Foundation grant CCF-2441912 (NSF CAREER), Air Force Office of Scientific Research under award number FA9550-25-1-0146, and the U.S. Department of Energy, Office of Advanced Scientific Computing Research under Award Number DE-SC0025430.

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

APPENDIX

## A  TABLE OF NOTATION

Table 5: Main symbols and notation used in the paper.

| Symbol | Meaning / Definition |
|---|---|
| *Domains, spaces, and operators* | |
| $D := (-L/2, L/2)^d$ | Periodic $d$-dimensional spatial box of side length $L$ |
| $\mathcal{G}$ | Ground-truth solution operator to be learned |
| *Spectral & spatial sampling* | |
| $\Xi = \{\boldsymbol{\xi}_1, \ldots, \boldsymbol{\xi}_m\}$ | Retained Fourier modes (truncated spectrum); $m = \lvert \Xi \rvert$ |
| $\mathcal{F}$ | Fourier transform |
| $\mathcal{F}_m$, $\mathcal{F}_m^{-1}$ | Truncated Fourier transform by $\Xi$ and its band-limited inverse |
| $\mathcal{Y} = \{\mathbf{y}_1, \ldots, \mathbf{y}_m\} \subset D$ | Uniform spatial grid |
| $h$ | Grid spacing of $\mathcal{Y}$ |
| *Fourier Neural Operator (FNO)* | |
| $\mathcal{L}_{\text{FNO}}$ | Single FNO layer |
| $\mathcal{G}_\theta^{\text{FNO}}$ | FNO network |
| $\mathbf{R}_\theta(\boldsymbol{\xi})$ | Learnable block-diagonal spectral multiplier |
| $\mathbf{W}_\theta$ | Point-wise learnable linear map |
| $\sigma(\cdot)$ | Point-wise non-linear activation |
| *KANO layer, symbol calculus, and projection* | |
| $\mathcal{L}_{\text{KANO}}$ | Single KANO layer |
| $\boldsymbol{p}(\mathbf{x}, \boldsymbol{\xi})$ | Learnable pseudo-differential symbol |
| $\mathbf{Op}_m(\boldsymbol{p})$ | Kohn-Nirenberg quantization of width $m$ defined by $\boldsymbol{p}$ |
| $\Pi_{\text{KANO}}(\mathcal{G})$ | KANO projection of $\mathcal{G}$ |
| $\mathcal{G}_\theta^{\text{KANO}}$ | KANO network |
| $\boldsymbol{\Phi}_\theta$ | Learnable activation |
| *Kolmogorov–Arnold Network (KAN) primitives* | |
| $\phi_{q,p}^{(\ell)}(\cdot)$ | 1D edge function on layer $\ell$, connecting $p^{\text{th}}$ node of layer $\ell$ to $q^{\text{th}}$ node of layer $(\ell + 1)$ |
| $b(t)$, $\{B_i(t)\}$ | Base function and B-spline basis used to parametrize $\phi_{q,p}^{(\ell)}(t)$ |
| *Q-KANO (quantum dynamics) notation* | |
| $\psi(x)$ | Input wave function |
| $w(x)$ | Quartic double-well potential |
| $\Delta T$ | Time step of propagation |
| $\boldsymbol{\phi}_\theta(\mathbf{x}, \boldsymbol{\xi})$ | Parametrized phase for symbol $\boldsymbol{p}(\mathbf{x}, \boldsymbol{\xi})$ of Q-KANO |
| $\boldsymbol{\vartheta}_\theta(\lvert \cdot \rvert, \angle \cdot)$ | Parametrized phase for non-linear activation of Q-KANO |
| $\mathcal{G}_\theta^{\text{Q-KANO}}$ | Q-KANO network |
| *Function spaces* | |
| $L^2(D)$ | Square-integrable function space on domain $D$ |
| $H^s(D)$ | Sobolev function space of order $s \geq 0$ on domain $D$ |

## B  EXPERIMENT DETAILS

### B.1  SYNTHETIC OPERATOR BENCHMARK

All experiments are carried out on periodic functions $f : \mathbb{T} \to \mathbb{R}$ with $\mathbb{T} = (-\pi, \pi]$ and a uniform trigonometric grid

$$x_j = x_{\min} + j\,\Delta x, \quad \Delta x = \frac{2\pi}{N}, \quad j = 0, \ldots, N-1,$$

with $N = 128$. Unless noted otherwise every random quantity is drawn *independently for every sample*.

**Outer envelope.** To avoid the Gibbs phenomenon all basis functions are multiplied by a smooth taper that decays to zero in a $\pi/6$-wide buffer near the periodic boundary:

$$A(x) = \begin{cases} 1, & |x| \leq 5\pi/6, \\ \cos^4\!\left[\dfrac{|x| - 5\pi/6}{\pi/6}\,\dfrac{\pi}{2}\right], & 5\pi/6 < |x| < \pi, \\ 0, & |x| \geq \pi. \end{cases}$$

The full "base" function is always $f_{\text{base}}(x) = A(x)\,g(x)$.

**Spectral derivatives and ground-truth operator.** Derivatives are computed with an exact Fourier stencil:

$$f'(x) \;=\; \boldsymbol{\mathcal{F}}^{-1}\!\big[\,i\xi\,\widehat{f}(\xi)\big], \qquad f''(x) \;=\; \boldsymbol{\mathcal{F}}^{-1}\!\big[-\xi^2\widehat{f}(\xi)\big].$$

$U[a,b]$ denotes random digit drawn from range $[a,b]$.

**Training families (Group A)**

**A1. sine_beats**:

$$g(x) = \sin(\omega_1 x + \phi_1)\,\sin(\omega_2 x + \phi_2), \quad \omega_i = 8\,U[0.5,3], \; \phi_i = U[0,2\pi].$$

**A2. chirped_cosine**:

$$g(x) = \cos\!\big(\alpha x^2\big), \qquad \alpha = 12\,U[0.5,2].$$

**A3. periodic** (random harmonic series):

$$g(x) = \sin(\omega x + \phi_1) + \cos(\omega x + \phi_2), \qquad \omega = 8\,U[0.5,3], \; \phi_{1,2} = U[0,2\pi].$$

**Unseen families (Group B)**

**B1. wave_packet**:

$$g(x) = \exp\!\left[-\frac{(x-\mu)^2}{2\sigma^2}\right]\sin\!\big(\omega x + \phi\big), \; \mu = U[-2,2], \; \sigma = \tfrac{1}{12}\,U[0.5,2], \; \omega = 12\,U[2,6], \; \phi = U[0,2\pi].$$

**B2. sinc_pulse**:

$$g(x) = \begin{cases} \dfrac{\sin(\alpha x)}{\alpha x}, & |x| > 10^{-12}, \\ 1, & |x| \leq 10^{-12}, \end{cases} \qquad \alpha = 12\,U[0.5,3].$$

**B3. gaussian_hermite**:

$$g(x) = H_n\!\Big(\frac{x-\mu}{\sigma}\Big)\,\exp\!\left[-\frac{(x-\mu)^2}{2\sigma^2}\right], \; n \in \{1,2,3\} \text{ uniform}, \; \mu = U[-2,2], \; \sigma = \tfrac{1}{8}\,U[0.5,2],$$

where $H_n$ is the degree-$n$ Hermite polynomial.

NORMALIZATION

Each realization is divided by its maximum absolute value, $\|f\|_\infty$, to obtain $\|f\|_\infty = 1$. The envelope guarantees periodicity and keeps the numerical spectrum sharply band-limited.

**Sample counts.** #train $= 2000$ samples from the three Group A families for train data and #test $= 400$ samples each from the Group A and Group B families for generalization tests.

## B.2 Quantum Dynamics Benchmark

We model a quantum apparatus with 200 state-preparation protocols each with perfect reproducibility, capable of generating an identical initial state whose wave function is drawn from one of the three families: *Periodic*, *Gaussian wave-packet*, and *Gaussian×Hermite*. The prepared initial states evolve under one of two unknown, time-independent Hamiltonians, and two arrays of 128 detectors measure position and momentum on uniform grids, yielding probability mass functions (PMFs) every 0.1ms for 100 time steps. PMFs collected from the first 10 time steps are used for training the models, and the rest of the PMFs collected from the remaining 90 time steps are used to evaluate the long-horizon fidelity drop beyond the train steps.

### B.2.1 Quantum Apparatus Assumptions

1. **State preparation.** A *collection of calibrated protocols* can each prepare a designated initial wave-function each of one of three real-valued families *Periodic*, *Gaussian wave-packet*, or *Gaussian–Hermite*: $\psi_0^{(m)}(x) \in L^2(\mathbb{T})^8$, $m = 1, \ldots, 200$. Repeated shots under the *same* protocol start from **exactly** the same $\psi_0^{(m)}$, enabling trajectory-level reproducibility for every member of the ensemble.

2. **Hamiltonian stability.** The (unknown) Hamiltonian is *time–independent*, so trajectories are perfectly repeatable once $\psi_0$ is fixed.

3. **Dual-basis detection.** Two 128-grid projective detectors measure the position basis $\{|x_i\rangle\}$ and the momentum basis $\{|\xi_j\rangle\}$, yielding empirical probability mass functions (PMFs) $\hat{p}_x(i) = |\psi(x_i)|^2$ and $\hat{p}_\xi(j) = |\widehat{\psi}(\xi_j)|^2$ on a common torus grid $\mathbb{T}_L$, $L = 4$.

### B.2.2 Data Generation Details

For each of 200 distinct sample trajectories we

1. draw the initial wave function and propagate on the Hamiltonian with a high-resolution Strang split: $\delta t = 1\,\mu\mathrm{s}$ for $10\,000$ micro-steps, producing coarse snapshots every 100 steps ($\Delta T = 0.1\,\mathrm{ms}$, $T = 1, \ldots, 100$);

2. store $(\psi_T,\, \mathbf{p}_x^T,\, \mathbf{p}_\xi^T)$ where $p_x^T(i) = |\psi_T(x_i)|^2$ and $p_\xi^T(j) = |\widehat{\psi}_T(\xi_j)|^2$.

Only the first 10 coarse steps are used for training; the remaining 90 steps test fidelity drop on long-horizon. All simulations employ an $n = 128$-point FFT grid to match the detectors.

**Spatial discretization.** We place the problem on a periodic box of length $L = 4$ with $N = 128$ grid points $x_j = x_{\min} + j\Delta x$, $\Delta x = L/N$. Periods suppress wrap–around artifacts because every initial state is tapered by the smooth envelope $A(x)$ defined in Appendix B.1. Spatial derivatives are taken spectrally: let $\xi_m = 2\pi m/L$ for $m = -N/2, \ldots, N/2 - 1$. Writing $\widehat{\psi}_m = \mathcal{F}[\psi](\xi_m)$,

$$\partial_x \psi = \mathcal{F}^{-1}[i\,\xi_m \widehat{\psi}_m], \qquad \partial_{xx}\psi = \mathcal{F}^{-1}[-\xi_m^2 \widehat{\psi}_m].$$

**Strang–splitting time integrator.** Let $\mathcal{K} := -\frac{1}{2}\partial_{xx}$ (kinetic), $\mathcal{V} : \psi \mapsto w(x) \cdot \psi$ (potential) and $\mathcal{N} : \psi \mapsto |\psi|^2 \cdot \psi$ (cubic nonlinearity). With time step $\Delta t$ the second–order Strang factorization reads

$$e^{(\mathcal{K}+\mathcal{V}+\mathcal{N})\Delta t} = e^{\frac{\Delta t}{2}(\mathcal{K}+\mathcal{V})} e^{\Delta t \mathcal{N}} e^{\frac{\Delta t}{2}(\mathcal{K}+\mathcal{V})} + \mathcal{O}(\Delta t^3).$$

Because $\mathcal{K}$ is diagonal in Fourier space and $\mathcal{V}$ in real space we implement each half step explicitly:

$$\psi \xleftarrow{\;\mathcal{V}/2\;} e^{-\frac{i}{2}\Delta t\, w(x)}\psi,$$

$$\widehat{\psi} \xleftarrow{\;\mathcal{K}/2\;} \mathcal{F}[\psi], \quad \widehat{\psi}_m \leftarrow e^{-\frac{i}{4}\Delta t\, \xi_m^2}\widehat{\psi}_m,$$

$$\psi \xleftarrow{\;\mathcal{N}\;} \mathcal{F}^{-1}[\widehat{\psi}], \quad \psi \leftarrow e^{-i\Delta t|\psi|^2}\psi,$$

repeat $\mathcal{K}/2$ and $\mathcal{V}/2$.

---

[8]Square-integrable function space.

The $L^2$ norm is renormalized after every macro step to compensate numerical drift. For DW Hamiltonian, non-linearity time evolution is omitted.

**Integrator parameters.** We use an inner step $\delta t = 10^{-6}$. A *macro* step of $\Delta T = 100 \, \delta t = 10^{-4}$ is recorded and the sequence is propagated for $T_{\max} = 100 \, \Delta T = 10^{-2}$.

**Initial–state families (real wave-functions).** Each sample draws a real profile $f(x)$ from $\{A_1, A_2, A_3\}$ below, multiplies it by the envelope $A$ and normalizes it in $L^2$:

$$\psi_0(x) = \frac{A(x) \, f(x)}{\|Af\|_2}.$$

**A1. random harmonic series**

$$f(x) = \sum_{m=0}^{5} \tfrac{1}{2} \xi_m^{(c)} \cos(2\pi m\xi) + \tfrac{1}{2} \xi_m^{(s)} \sin(2\pi m\xi), \quad \xi = \frac{x+2}{4}, \ \xi_m^{(\cdot)} \sim \mathcal{N}(0,1).$$

**A2. Gaussian wave packet** $f(x) = \exp[-(x - x_0)^2/(2\sigma^2)]$ with $x_0 \sim U[-0.4\pi, 0.4\pi]$ and $\sigma \sim U[0.1, 0.3]$.

**A3. Gaussian–Hermite mode** $f(x) = H_n\left(\frac{x-x_0}{\sigma}\right) \exp[-(x - x_0)^2/(2\sigma^2)]$ with $n \in \{0, 1, 2\}$ uniform, $x_0, \sigma$ as above, and $H_n$ the Hermite polynomial.

**Stored quantities.** For every sample index $s$ and every snapshot $T \in \{0, 1, \ldots, 100\}$ we save

$$\big(\mathbf{x}, \ \psi(x,T), \ |\psi(x,T)|^2, \ |\widehat{\psi}(\xi,T)|^2\big) \quad \longrightarrow \quad \texttt{wavefunc, pos\_pdf, mom\_pdf.}$$

All arrays are written in `float32` except the complex wave-function, stored as `complex64`. Altogether one call to `generate_dataset(num_samples=200)` produces $200 \times 101 \times 4 = 80{,}800$ labeled records.

### B.2.3 TRAIN TYPE DETAILS

Let $\psi_{\Theta}^{(T)}$ be the $T$-step prediction of Q-KANO given $\psi_0$.

| Train Type | Train Dataset | Loss function |
|---|---|---|
| **full** | complex $\psi^{(T)}$ | $\mathcal{L}_{full} = \dfrac{\|\psi_{\Theta}^{(T)} - \psi^{(T)}\|_2}{\|\psi^{(T)}\|_2}$ |
| **pos** | $\mathbf{p}_x^{(T)}$ | $\mathcal{L}_{pos} = D_{\mathrm{KL}}\big(\mathbf{p}_x^{(T)} \, \| \, |\psi_{\Theta}^{(T)}|^2\big)$ |
| **pos & mom** | $\mathbf{p}_x^{(T)}, \mathbf{p}_\xi^{(T)}$ | $\mathcal{L}_{pos\&mom} = D_{\mathrm{KL}}\big(\mathbf{p}_x^{(T)} \, \| \, |\psi_{\Theta}^{(T)}|^2\big) + D_{\mathrm{KL}}\big(\mathbf{p}_\xi^{(T)} \, \| \, |\widehat{\psi}_{\Theta}^{(T)}|^2\big)$ |

## C  PROOFS

### C.1  NOTATION AND PRELIMINARIES

Throughout, $\mathbb{T}^d := [-\pi, \pi]^d$ denotes the flat $d$–torus and $\mathbb{Z}^d$ the lattice of Fourier indices. For $\boldsymbol{\xi} \in \mathbb{Z}^d$ let $e_{\boldsymbol{\xi}}(\mathbf{x}) := e^{i\boldsymbol{\xi} \cdot \mathbf{x}}$. The Fourier coefficient of a square integrable function $\mathbf{f}$ is

$$\widehat{\mathbf{f}}(\boldsymbol{\xi}) := (2\pi)^{-d} \int_{\mathbb{T}^d} \mathbf{f}(\mathbf{x}) \, e^{-i\boldsymbol{\xi} \cdot \mathbf{x}} \, d\mathbf{x}.$$

In a Sobolev space $H^s$ with an order of smoothness $s \in \mathbb{R}$, the Sobolev norm of function $\mathbf{f}$ is

$$\|\mathbf{f}\|_{H^s}^2 := \frac{(2\pi)^d}{2} \sum_{\boldsymbol{\xi} \in \mathbb{Z}^d} (1 + |\boldsymbol{\xi}|^{2s}) \, |\widehat{\mathbf{f}}(\boldsymbol{\xi})|^2.$$

**Asymptotics.** Write $A \lesssim B$ if $A \leq C\,B$ for a constant $C$ depending only on fixed parameters (dimension, regularity exponents, etc.).

**Vector Notation** Fix a spatial dimension $d \geq 1$ and an index $j \in \{1, \ldots, d\}$. For $\boldsymbol{\xi} = (\xi_1, \ldots, \xi_d) \in \mathbb{Z}^d$ we write

$$\boldsymbol{\xi}_{-j} := (\xi_1, \ldots, \xi_{j-1}, \xi_{j+1}, \ldots, \xi_d) \in \mathbb{Z}^{d-1}$$

for the vector obtained by *removing* the $j$-th coordinate of $\boldsymbol{\xi}$. Conversely, for $\boldsymbol{\alpha} = (\alpha_1, \ldots, \alpha_{d-1}) \in \mathbb{Z}^{d-1}$ and $n \in \mathbb{Z}$ we define the *insertion* map

$$(\boldsymbol{\alpha}, n)_j := (\alpha_1, \ldots, \alpha_{j-1}, n, \alpha_j, \ldots, \alpha_{d-1}) \in \mathbb{Z}^d.$$

We use $|\cdot|_\infty$ for the max–norm on $\mathbb{Z}^d$, i.e. $|\boldsymbol{\xi}|_\infty = \max_{1 \leq i \leq d} |\xi_i|$. When we write $\widehat{\mathbf{u}}(\boldsymbol{\alpha}, r)$, this is shorthand for the $d$–dimensional coefficient $\widehat{\mathbf{u}}\big((\boldsymbol{\alpha}, r)_j\big)$.

## C.2  PROOF OF LEMMA 1

**Restatement of Lemma 1 (with explicit notation).** Let $\mathbf{u} \in H^s(\mathbb{T}^d)$ with $s > 0$, and assume that its Fourier coefficients are compactly supported:

$$\widehat{\mathbf{u}}(\boldsymbol{\xi}) = \mathbf{0} \quad \text{for all } |\boldsymbol{\xi}|_\infty > N_0, \qquad \text{and} \qquad \widehat{\mathbf{u}} \not\equiv \mathbf{0}.$$

Fix $j \in \{1, \ldots, d\}$ and set $\mathbf{v}(\mathbf{x}) := x_j \, \mathbf{u}(\mathbf{x})$. Then there exist

$$\boldsymbol{\alpha} \in \mathbb{Z}^{d-1}, \qquad m \in \{1, 2, \ldots, 2N_0 + 1\}, \qquad c > 0, \quad R \in \mathbb{N},$$

and an infinite set of the fiber at $j^{\text{th}}$ coordinate,

$$\Xi_{\boldsymbol{\alpha}, R} \subset \big\{ \boldsymbol{\xi} \in \mathbb{Z}^d : \boldsymbol{\xi}_{-j} = \boldsymbol{\alpha}, \, |\xi_j| \geq R \big\}$$

such that

$$\big|\widehat{\mathbf{v}}(\boldsymbol{\xi})\big| \geq \frac{c}{\big(1 + |\xi_j|\big)^m} \qquad \forall \boldsymbol{\xi} \in \Xi_{\boldsymbol{\alpha}, R}. \tag{20}$$

In particular, if for some $\boldsymbol{\alpha} \in \mathbb{Z}^{d-1}$ one has

$$\sum_{r=-N_0}^{N_0} \widehat{\mathbf{u}}\big((\boldsymbol{\alpha}, r)_j\big) \neq \mathbf{0} \quad \text{(equivalently,} \quad \sum_{r=-N_0}^{N_0} \widehat{\mathbf{u}}(\boldsymbol{\alpha}, r) \neq \mathbf{0} \text{ in the shorthand above),}$$

then the bound equation 20 holds with the sharper exponent $m = 1$.

*Proof.* We work with square-integrable function $\mathbf{v} \in L^2(\mathbb{T}^d)$ where $\mathbb{T}^d = [0, 2\pi]^d$.

**Step 1 — The periodic "coordinate" and its Fourier coefficients.** The function $\mathbf{x} \mapsto x_j$ is not periodic. Introduce the zero-mean, periodic 1D sawtooth

$$\psi_j(\mathbf{x}) := x_j - \pi, \qquad \mathbf{x} \in [0, 2\pi]^d,$$

extended periodically to $\mathbb{T}^d$. A direct computation (factorization of the integral and one-dimensional integration by parts) shows that its Fourier coefficients are supported on the $j^{\text{th}}$-coordinate: for $\boldsymbol{\xi} \in \mathbb{Z}^d$,

$$\widehat{\psi_j}(\boldsymbol{\xi}) = \begin{cases} -\dfrac{1}{i\,\xi_j}, & \text{if } \boldsymbol{\xi}_{-j} = \mathbf{0} \text{ and } \xi_j \neq 0, \\ 0, & \text{if } \boldsymbol{\xi} = \mathbf{0} \text{ or } \boldsymbol{\xi}_{-j} \neq \mathbf{0}. \end{cases} \tag{21}$$

Moreover $x_j = \psi_j + \pi$, hence

$$\widehat{\mathbf{v}}(\boldsymbol{\xi}) = \mathcal{F}[\psi_j \mathbf{u}](\boldsymbol{\xi}) + \pi \, \widehat{\mathbf{u}}(\boldsymbol{\xi}). \tag{22}$$

**Step 2 — Exact coefficient formula outside the support of $\widehat{\mathbf{u}}$.** Since $\widehat{\mathbf{u}}(k) = \mathbf{0}$ for $|\boldsymbol{\xi}|_\infty > N_0$, the second term in equation 22 vanishes whenever $|\boldsymbol{\xi}|_\infty > N_0$. Using equation 21 and the convolution theorem, we obtain for any $\boldsymbol{\xi} \in \mathbb{Z}^d$ with $|\boldsymbol{\xi}|_\infty > N_0$:

$$\widehat{\mathbf{v}}(\boldsymbol{\xi}) = \sum_{\boldsymbol{\ell} \in \mathbb{Z}^d} \widehat{\psi_j}(\boldsymbol{\xi} - \boldsymbol{\ell}) \, \widehat{\mathbf{u}}(\boldsymbol{\ell}) = -\frac{1}{i} \sum_{\substack{\boldsymbol{\ell}_{-j} = \boldsymbol{\xi}_{-j} \\ |\ell_j| \leq N_0}} \frac{\widehat{\mathbf{u}}(\boldsymbol{\ell})}{\xi_j - \ell_j}. \tag{23}$$

Thus, along any fixed transverse index $\boldsymbol{\alpha} := \boldsymbol{\xi}_{-j} \in \mathbb{Z}^{d-1}$, the tail $\widehat{\mathbf{v}}(\boldsymbol{\alpha}, n)$ for large $|n|$ is a finite sum of simple fractions in the single variable $n$.

**Step 3 — Moments on a nontrivial fiber and the first non-vanishing moment.** Because $\widehat{\mathbf{u}} \not\equiv \mathbf{0}$, there exists at least one $\boldsymbol{\alpha} \in \mathbb{Z}^{d-1}$ for which the fiber

$$\mathcal{F}_{\boldsymbol{\alpha}} := \{r \in \mathbb{Z} : \ \widehat{\mathbf{u}}(\boldsymbol{\alpha}, r) \neq \mathbf{0}\}$$

is nonempty. Define the (finite) coefficients $\mathbf{c}_r := \widehat{\mathbf{u}}(\boldsymbol{\alpha}, r)$ for $r \in [-N_0, N_0]$, and their moments

$$\boldsymbol{\mu}_p := \sum_{r=-N_0}^{N_0} r^p \mathbf{c}_r \qquad (p \geq 0).$$

Let $m \in \{1, \dots, 2N_0 + 1\}$ be the smallest index for which $\boldsymbol{\mu}_{m-1} \neq \mathbf{0}$. Such an $m$ exists since not all $\mathbf{c}_r$ vanish.

**Step 4 — Asymptotics along a line and a polynomial lower bound.** For $n \in \mathbb{Z}$ with $|n| > N_0$, formula equation 23 along the line $\boldsymbol{\xi}_{-j} = \boldsymbol{\alpha}$ reads

$$\widehat{\mathbf{v}}(\boldsymbol{\alpha}, n) = -\frac{1}{i} \sum_{r=-N_0}^{N_0} \frac{\mathbf{c}_r}{n - r}.$$

Expanding $\frac{1}{n-r} = \frac{1}{n} \sum_{q \geq 0} \left(\frac{r}{n}\right)^q$ for $|n| > 2N_0$ and collecting terms yields the asymptotic expansion

$$\widehat{\mathbf{v}}(\boldsymbol{\alpha}, n) = -\frac{1}{i} \left( \frac{\boldsymbol{\mu}_0}{n} + \frac{\boldsymbol{\mu}_1}{n^2} + \cdots + \frac{\boldsymbol{\mu}_{m-1}}{n^m} + O\left( \frac{1}{|n|^{m+1}} \right) \right), \qquad |n| \to \infty.$$

By the choice of $m$, the first nonzero term is $\boldsymbol{\mu}_{m-1}/n^m$. Consequently, there exist $R \in \mathbb{N}$ and $c > 0$ such that

$$\left| \widehat{\mathbf{v}}(\boldsymbol{\alpha}, n) \right| \geq \frac{c}{|n|^m} \qquad \text{for all } |n| \geq R, \ n \in \mathbb{Z}.$$

**Step 5 — Conclusion and the special case** $m = 1$. Then equation 20 holds for all $\boldsymbol{\xi} \in \Xi_{\boldsymbol{\alpha}, R}$ with the exponent $m$ determined in Step 3, by the definition of $\Xi_{\boldsymbol{\alpha}, R}$ and that $|n| \geq R$. If $\boldsymbol{\mu}_0 = \sum_r \mathbf{c}_r \neq \mathbf{0}$ for the chosen fiber (equivalently, $\sum_{r=-N_0}^{N_0} \widehat{\mathbf{u}}(\boldsymbol{\alpha}, r) \neq \mathbf{0}$), then $m = 1$ and we obtain the sharper $|\widehat{\mathbf{v}}(\boldsymbol{\xi})| \gtrsim (1 + |\xi_j|)^{-1}$ along the line $\boldsymbol{\xi}_{-j} = \boldsymbol{\alpha}$. $\qquad \square$

**Remark 4.** *The explicit one-dimensional tail equation 23 shows that multiplying a band-limited field by the coordinate $x_j$ produces a* polynomial *Fourier tail decay along lines parallel to the $j^{th}$ coordinate, with rate $(1 + |\xi_j|)^{-m}$ where $m$ is the first non-vanishing moment of the finitely many coefficients on the relevant fiber. In particular, when $m = 1$, the decay is exactly $(1 + |\xi_j|)^{-1}$. Such algebraic tails are consistent with the pseudo-spectral projection error estimate quoted in Kovachki et al. (2021, Thm. 40).*

### C.3 PROOF OF THEOREM 1

**Restatement of Theorem 1.** Let $\boldsymbol{\alpha} = (\alpha_1, \dots, \alpha_d) \in \mathbb{N}^d$ with total degree $M := |\boldsymbol{\alpha}| \geq 1$ and define the position-multiplier

$$\mathcal{M}(\mathbf{x}) := x_1^{\alpha_1} x_2^{\alpha_2} \cdots x_d^{\alpha_d}.$$

For inputs band-limited to radius $N_0$ and lying in $H^s(\mathbb{T}^d)$ with $s > \frac{d}{2}$, any Fourier Neural Operator $\mathcal{G}_\theta^{\mathrm{FNO}}$ that achieves $\|\mathcal{M}(\mathbf{x}) - \mathcal{G}_\theta^{\mathrm{FNO}}\|_{H^s \to H^{s-\delta}} \leq \varepsilon$ ($0 < \delta < 1$) must employ a spectral bandwidth (FNO width) $m \gtrsim \varepsilon^{-M/s}$ and a parameter count $|\theta| \geq \exp\left(c \, \varepsilon^{-Md/s}\right)$ for some $c > 0$ depending only on $(d, s, \delta, N_0)$.

*Proof.* Set $s' := s - \delta$ with $0 < \delta < 1$.

**Step 1 – Algebraic tail produced by $\mathcal{M}(\mathbf{x})$.** Applying Lemma 1 once per factor of $x_j$ shows that for some constant $C_0 > 0$ and an infinite set $\Xi_\infty \subset \mathbb{Z}^d$,

$$|\mathcal{F}[\mathcal{M}(\mathbf{x})\mathbf{u}](\boldsymbol{\xi})| \geq \frac{C_0}{(1 + |\boldsymbol{\xi}|)^{M+1}} \qquad \forall \boldsymbol{\xi} \in \Xi_\infty. \tag{24}$$

**Step 2 – Pseudo-spectral projection lower bound.** For any $\mathbf{f} \in H^s$, pseudo-spectral projection error estimate gives (Gottlieb & Orszag, 1977)

$$\|(\mathcal{I} - \mathbf{\Pi}_{\text{FNO}})\mathbf{f}\|_{H^{s'}} \ \geq \ C_1 \Big[ \sum_{|\boldsymbol{\xi}|_\infty > N} (1 + |\boldsymbol{\xi}|^{2s'}) \, |\widehat{\mathbf{f}}(\boldsymbol{\xi})|^2 \Big]^{1/2},$$

where $\mathcal{I}$ is an identity operator. Insert $\mathbf{f} = \boldsymbol{\mathcal{M}}(\mathbf{x})\mathbf{u}$ and the tail bound equation 24; summing over $\Xi_\infty \cap \{|\boldsymbol{\xi}|_\infty > m\}$ yields

$$\|(\mathcal{I} - \mathbf{\Pi}_{\text{FNO}})\boldsymbol{\mathcal{M}}(\mathbf{x})\mathbf{u}\|_{H^{s'}} \ \gtrsim \ m^{-(M-\delta)}.$$

Imposing this residual $\leq \frac{1}{2}\varepsilon$ forces

$$m \ \geq \ C_2 \, \varepsilon^{-M/s}, \qquad C_2 = C_2(d, s, \delta, M, N_0) > 0. \tag{25}$$

**Step 3 – Canonical neural scaling in the latent map.** An FNO with spectral radius (width) $m$ manipulates a latent vector of dimension $(2m+1)^d \sim m^d$. Approximating a generic Lipschitz map $\mathbf{G} : \mathbb{C}^{m^d} \to \mathbb{C}^{m^d}$ to accuracy $\varepsilon/2$ with a fully connected network requires (Yarotsky, 2017; De Ryck et al., 2021) *at least*

$$\text{parameters} \ \gtrsim \ \varepsilon^{-m^d}.$$

(Kovachki et al., 2021, Remark 22)

**Step 4 – Substitute the bandwidth lower bound.** Using equation 25, $m^d \sim \varepsilon^{-Md/s}$. Hence the latent network must have at least

$$|\theta| \ \gtrsim \ \varepsilon^{-\varepsilon^{-Md/s}},$$

a *super-exponential* curse of dimensionality in the target accuracy $\varepsilon$.

$\square$

### C.4 PROOF OF THEOREM 2

**Restatement of Theorem 2.** Let $s \geq s' \geq 0$ and $s_p > d/2$. Consider any finite composition

$$\boldsymbol{\mathcal{G}} \ = \ (\varsigma_\ell \circ \boldsymbol{\mathcal{L}}_\ell) \circ \cdots \circ (\varsigma_1 \circ \boldsymbol{\mathcal{L}}_1), \qquad \boldsymbol{\mathcal{L}}_i = \mathbf{Op}_m(\boldsymbol{p}_i),$$

where each Kohn–Nirenberg symbol $\boldsymbol{p}_i$ belongs to $W^{s_p,2}\big(\mathbb{T}_{\mathbf{x}}^d \times \mathbb{T}_{\boldsymbol{\xi}}^d\big)^9$, and each $\varsigma_i$ is a point-wise non-linearity of uniformly bounded degree (so the number of such nonlinearities is $O(\ell)$ and independent of $\varepsilon$). For every $\varepsilon > 0$ there exists a *single-layer* KANO $\boldsymbol{\mathcal{G}}_\theta^{\text{KANO}}$ such that

$$\|\boldsymbol{\mathcal{G}} - \boldsymbol{\mathcal{G}}_\theta^{\text{KANO}}\|_{H^s \to H^{s'}} \leq \varepsilon, \qquad |\theta| \ = \ O\big(\varepsilon^{-d/(2s_p)}\big).$$

*Proof.* **Step 1 – Kohn–Nirenberg quantization error.** By the quadrature bound from Demanet & Ying (2011), for each $\boldsymbol{\mathcal{L}}_i$ we have

$$\big\|\boldsymbol{\mathcal{L}}_i - \mathbf{Op}_m(\boldsymbol{p}_i)\big\|_{H^s \to H^{s'}} \ \leq \ C' \, B \, m^{-(s-s')},$$

where $\mathbf{Op}_m(\boldsymbol{p}_i)$ keeps only frequencies $|\boldsymbol{\xi}|_\infty \leq m$. Since each $\varsigma_i$ is a bounded-degree pointwise map, its Nemytskii operator is Lipschitz on bounded sets; write $L_i := \text{Lip}(\varsigma_i)^{10}$ on the relevant range and set $L_{\max} := \max_i L_i$. Let $M := \max_j\{\|\boldsymbol{\mathcal{L}}_j\|_{H^{s'} \to H^{s'}}, \|\mathbf{Op}_m(\boldsymbol{p}_j)\|_{H^{s'} \to H^{s'}}\}$. Because $s_p > \frac{d}{2}$ implies $\|\boldsymbol{p}_j\|_{L^\infty} \lesssim \|\boldsymbol{p}_j\|_{W^{s_p,2}}$, both $\boldsymbol{\mathcal{L}}_j$ and $\mathbf{Op}_m(\boldsymbol{p}_j)$ are bounded on $H^{s'}$ with a bound independent of $m$. A telescoping estimate for the interleaved composition then yields

$$\big\|\boldsymbol{\mathcal{G}} - \mathbf{\Pi}_{\text{KANO}}(\boldsymbol{\mathcal{G}})\big\|_{H^s \to H^{s'}} \ \leq \ C_* \, C' \, B \, m^{-(s-s')}, \qquad C_* \ \leq \ \ell \, (L_{\max} M)^{\ell-1} L_{\max}.$$

---

[9] Standard square integrable periodic Sobolev space on the product torus $\mathbb{T}^{2d}$.

[10] Lipschitz constant of the point-wise nonlinearity $\varsigma_i$ on the relevant value range, i.e. $L_i :=$ $\sup_{a \neq b} \frac{|\varsigma_i(a) - \varsigma_i(b)|}{|a-b|}$ with $a, b$ restricted to the compact interval attained by the $i$-th preactivations.

Pick $m := \left(2\,C_*\,C'B/\varepsilon\right)^{1/(s-s')}$. Then the full composition $\mathcal{G}$ deviates from its projected counterpart

$$\mathbf{\Pi}_{\mathrm{KANO}}(\mathcal{G}) := (\varsigma_\ell \circ \mathbf{Op}_m(\boldsymbol{p}_\ell)) \circ \cdots \circ (\varsigma_1 \circ \mathbf{Op}_m(\boldsymbol{p}_1))$$

by at most $\varepsilon/2$ in $H^s \to H^{s'}$ operator norm.

**Step 2 – KAN approximation of both symbols and pointwise nonlinearities.** By the width-fixed KAN scaling law from Wang et al. (2024), for any $\eta > 0$ there exists a Kolmogorov–Arnold Network $\boldsymbol{\Phi}_{i,\eta}$ with $|\boldsymbol{\Phi}_{i,\eta}| = O(\eta^{-d/(2s_p)})$ such that $\|\boldsymbol{p}_i - \boldsymbol{\Phi}_{i,\eta}\|_{L^\infty} \leq \eta$ on $\mathbb{T}_{\mathbf{x}}^d \times \mathbb{T}_{\boldsymbol{\xi}}^d$. Likewise, since each $\varsigma_i$ is a fixed-degree pointwise map, its action over the compact value range visited by the projected flow can be uniformly approximated by a width-fixed KAN $\boldsymbol{\Psi}_{i,\eta}$ with size $|\boldsymbol{\Psi}_{i,\eta}| = O(\eta^{-1/(2s_p)})$ and $\|\varsigma_i - \boldsymbol{\Psi}_{i,\eta}\|_{L^\infty} \leq \eta$. Choose

$$\eta_{\mathrm{sym}} := \frac{\varepsilon}{4\ell}, \qquad \eta_{\mathrm{nl}} := \frac{\varepsilon}{4\ell}.$$

Define the single-layer KANO

$$\mathcal{G}_\theta^{\mathrm{KANO}} := (\boldsymbol{\Psi}_{\ell,\eta_{\mathrm{nl}}} \circ \mathbf{Op}_m(\boldsymbol{\Phi}_{\ell,\eta_{\mathrm{sym}}})) \circ \cdots \circ (\boldsymbol{\Psi}_{1,\eta_{\mathrm{nl}}} \circ \mathbf{Op}_m(\boldsymbol{\Phi}_{1,\eta_{\mathrm{sym}}})),$$

where the same spectral radius $m$ from Step 1 is used in every $\mathbf{Op}_m(\,\cdot\,)$.

**Step 3 – Error accumulation beyond projection.** Using linearity of the symbol-to-operator map and stability of Nemytskii (pointwise) maps under uniform approximation, the post-projection error splits into a sum of the symbol parts and the nonlinearity parts:

$$\left\|\mathbf{\Pi}_{\mathrm{KANO}}(\mathcal{G}) - \mathcal{G}_\theta^{\mathrm{KANO}}\right\|_{H^s \to H^{s'}} \leq \sum_{i=1}^{\ell} \left\|\mathbf{Op}_m(\boldsymbol{p}_i) - \mathbf{Op}_m(\boldsymbol{\Phi}_{i,\eta_{\mathrm{sym}}})\right\|_{H^s \to H^{s'}} + \sum_{i=1}^{\ell} \|\varsigma_i - \boldsymbol{\Psi}_{i,\eta_{\mathrm{nl}}}\|_{L^\infty}$$

$$\leq \ell\,\eta_{\mathrm{sym}} + \ell\,\eta_{\mathrm{nl}} \;=\; \varepsilon/2.$$

Combining with Step 1 yields $\|\mathcal{G} - \mathcal{G}_\theta^{\mathrm{KANO}}\|_{H^s \to H^{s'}} \leq \varepsilon$.

**Step 4 – Parameter complexity.** Summing the sizes of all KAN blocks gives

$$|\theta| \;=\; \sum_{i=1}^{\ell} O(\eta_{\mathrm{sym}}^{-d/(2s_p)}) \;+\; \sum_{i=1}^{\ell} O(\eta_{\mathrm{nl}}^{-1/(2s_p)}) \;+\; O(1) \;=\; O\big((\varepsilon/\ell)^{-d/(2s_p)}\big) + O\big((\varepsilon/\ell)^{-1/(2s_p)}\big).$$

Since $\ell$ and the number/degree of the $\varsigma_i$ are fixed (do not scale with $\varepsilon$), the dominating term is $O(\varepsilon^{-d/(2s_p)})$, establishing the claimed complexity. $\qquad\square$

# D    ADDITIONAL DETAILS ON THE PURE-SPECTRAL BOTTLENECK OF FNO

In Section 3.1 we performed a linear analysis of a single FNO layer via its Jacobian to illustrate the pure-spectral bottleneck. We showed that, in the first-order approximation, all spectral off–diagonals arise from the spectrum of the input–dependent gate $\sigma'\big(\mathbf{z}(\mathbf{u}, \cdot)\big)$ (12). Hence, although a single FNO layer is capable of generating spectral off-diagonals, they are tied to the input distribution and leads to the structural fragility in out-of-distribution performance (generalization on the unseen input distribution). Experimental results in Section 5 are aligned with our concerns.

In this Appendix, we show that the same phenomenon persists at *every* order of the Fréchet expansion of a single FNO layer, and composing deep layers does not remove the input distribution dependence of the spectral off-diagonals generated by the model as well.

Throughout, we work on the flat torus $\mathbb{T}^d = [-\pi, \pi]^d$ and Sobolev spaces $H^s$ as in Appendix C.

## D.1    HIGHER–ORDER SPECTRAL OFF–DIAGONALS OF FNO

Recall the FNO layer $\mathcal{L}_{\mathrm{FNO}}$ (4):

$$\mathcal{L}_{\mathrm{FNO}}(\mathbf{u})(\mathbf{x}) \;=\; \sigma\Big(\mathcal{F}_m^{-1}\Big(\mathbf{R}_{\theta_1}(\boldsymbol{\xi}) \cdot \mathcal{F}_m(\mathbf{u})(\boldsymbol{\xi})\Big)(\mathbf{x}) \;+\; \mathbf{W}_{\theta_2} \cdot \mathbf{u}(\mathbf{x})\Big).$$

For the analysis it is convenient to collect the linear terms into a linear operator

$$\boldsymbol{\mathcal{A}} \;:=\; \boldsymbol{\mathcal{F}}_m^{-1} \circ \mathbf{R}_\theta(\boldsymbol{\xi}) \circ \boldsymbol{\mathcal{F}}_m \;+\; \mathbf{W}_\theta, \tag{26}$$

so that

$$\boldsymbol{\mathcal{L}}_{\text{FNO}}(\mathbf{u})(\mathbf{x}) \;=\; \sigma\big(\mathbf{z}(\mathbf{u}, \mathbf{x})\big), \qquad \mathbf{z}(\mathbf{u}, \mathbf{x}) := (\boldsymbol{\mathcal{A}}\mathbf{u})(\mathbf{x}), \tag{27}$$

where $\sigma$ is the non-linear activation acting point-wise.

We first compute the $k$-th Fréchet derivative (Fréchet, 1906) of $\boldsymbol{\mathcal{L}}_{\text{FNO}}(\mathbf{u})(\mathbf{x})$.

**Proposition 1 (Structure of higher–order derivatives of a single Fourier layer).** *Let $\mathbf{u} \in H^s$ and $k$ be the order of the Fréchet derivative. Also let $\mathbf{h}_1, \dots, \mathbf{h}_k \in H^s$ be the arbitrary direction functions of each order. Then for every integer $k \geq 1$, the Fréchet derivative of $\boldsymbol{\mathcal{L}}_{\text{FNO}}(\mathbf{u})(\mathbf{x})$ is:*

$$\boldsymbol{\mathcal{D}}^k \boldsymbol{\mathcal{L}}_{\text{FNO}}(\mathbf{u})[\mathbf{h}_1, \dots, \mathbf{h}_k](\mathbf{x}) \;:=\; \sigma^{(k)}\big(\mathbf{z}(\mathbf{u}, \mathbf{x})\big) \prod_{j=1}^{k} (\boldsymbol{\mathcal{A}}\mathbf{h}_j)(\mathbf{x}), \tag{28}$$

*where $\sigma^{(k)}$ is the usual scalar $k$-th derivative. More explicitly, for each channel $q$,*

$$\big[\boldsymbol{\mathcal{D}}^k \boldsymbol{\mathcal{L}}_{\text{FNO}}(\mathbf{u})[\mathbf{h}_1, \dots, \mathbf{h}_k]\big]_q(\mathbf{x}) \;:=\; \sigma^{(k)}\big(\mathbf{z}_q(\mathbf{u}, \mathbf{x})\big) \prod_{j=1}^{k} [\boldsymbol{\mathcal{A}}\mathbf{h}_j]_q(\mathbf{x}). \tag{29}$$

*Proof.* Since $\boldsymbol{\mathcal{A}}$ (26) is linear, $\boldsymbol{\mathcal{D}}\mathbf{z}(\mathbf{u}) \equiv \boldsymbol{\mathcal{A}}$ and $\boldsymbol{\mathcal{D}}^r \mathbf{z}(\mathbf{u}) \equiv 0$ for all $r \geq 2$. By the chain rule,

$$\boldsymbol{\mathcal{D}}\boldsymbol{\mathcal{L}}_{\text{FNO}}(\mathbf{u})[\mathbf{h}_i](\mathbf{x}) \;=\; \sigma'\big(\mathbf{z}(\mathbf{u}, \mathbf{x})\big) \, (\boldsymbol{\mathcal{A}}\mathbf{h}_i)(\mathbf{x}),$$

where $\sigma'$ acts point-wise on each channel. For higher order derivatives, we apply the Faà di Bruno formula (Faa di Bruno, 1855). Because all higher derivatives of $\mathbf{z}$ vanish, every term involving $\boldsymbol{\mathcal{D}}^r \mathbf{z}(\mathbf{u})$ with $r \geq 2$ drops out, leaving

$$\boldsymbol{\mathcal{D}}^k \boldsymbol{\mathcal{L}}_{\text{FNO}}(\mathbf{u})[\mathbf{h}_1, \dots, \mathbf{h}_k] \;=\; \sigma^{(k)}(\mathbf{z}(\mathbf{u}))\big[\boldsymbol{\mathcal{A}}\mathbf{h}_1, \dots, \boldsymbol{\mathcal{A}}\mathbf{h}_k\big]. \tag{30}$$

Since $\sigma$ acts point-wise, this reduces to Eq. (28), and to Eq. (equation 29) on each channel. $\quad\square$

Thus, as apparent in Eq. (29), *every* Fréchet derivative is a point-wise product of

- an input-dependent gate $\sigma^{(k)}\big(\mathbf{z}(\mathbf{u}, \mathbf{x})\big)$ depending on the current input $\mathbf{u}$, and
- the product of $k$ fixed linear responses $(\boldsymbol{\mathcal{A}}\mathbf{h}_i)(\mathbf{x})$ incapable of generating spectral off-diagonals.

Therefore, the structure of the high-order derivatives of a single layer FNO is just as tied to the input $\mathbf{u}$ distribution as the first-order Jacobian shown in Section 3.1.

D.2 SPECTRAL REPRESENTATION OF HIGH-ORDER DERIVATIVES AND MULTI-LAYER FNO

Fix $k \geq 1$ and define

$$\mathbf{s}_k(\mathbf{u}, \mathbf{x}) := \sigma^{(k)}\big(\mathbf{z}(\mathbf{u}, \mathbf{x})\big), \qquad \mathbf{b}_i(\mathbf{x}) := (\boldsymbol{\mathcal{A}}\mathbf{h}_i)(\mathbf{x}), \quad i = 1, \dots, k. \tag{31}$$

Then Eq. (28) reads

$$\boldsymbol{\mathcal{D}}^k \boldsymbol{\mathcal{L}}_{\text{FNO}}(\mathbf{u})[\mathbf{h}_1, \dots, \mathbf{h}_k](\mathbf{x}) \;=\; \mathbf{s}_k(\mathbf{u}, \mathbf{x}) \cdot \prod_{i=1}^{k} \mathbf{b}_i(\mathbf{x}). \tag{32}$$

The Fourier transform of a product of $(k+1)$ functions is a $(k+1)$-fold convolution, so for each output frequency $\boldsymbol{\xi} \in \mathbb{Z}^d$ we have

$$\widehat{\boldsymbol{\mathcal{D}}^k \boldsymbol{\mathcal{L}}_{\text{FNO}}(\mathbf{u})[\mathbf{h}_1, \dots, \mathbf{h}_k]}(\boldsymbol{\xi}) \;=\; \big(\widehat{\mathbf{s}_k(\mathbf{u})} * \widehat{\mathbf{b}_1} * \cdots * \widehat{\mathbf{b}_k}\big)(\boldsymbol{\xi}). \tag{33}$$

Spectrally, the linear operator $\boldsymbol{\mathcal{A}}$ (26) acts diagonally:

$$\widehat{\boldsymbol{\mathcal{A}}\mathbf{h}_i}(\boldsymbol{\xi}) \;=\; \mathbf{A}(\boldsymbol{\xi}) \, \widehat{\mathbf{h}_i}(\boldsymbol{\xi}), \tag{34}$$

where $\mathbf{A}(\boldsymbol{\xi}) = \mathbf{R}(\boldsymbol{\xi}) + \mathbf{W}$ as in Eq. (12). Expanding the convolution in Eq. (33) yields

$$\mathcal{D}^k \mathcal{L}_{\text{FNO}}\widehat{(\mathbf{u})}[\mathbf{h}_1, \dots, \mathbf{h}_k](\boldsymbol{\xi}) = \sum_{\boldsymbol{\xi}_1, \dots, \boldsymbol{\xi}_k \in \mathbb{Z}^d} \underbrace{\widehat{\mathbf{s}_k(\mathbf{u})}(\boldsymbol{\xi} - \boldsymbol{\xi}_1 - \dots - \boldsymbol{\xi}_k)}_{\text{depends on input} \mathbf{u}} \cdot \prod_{i=1}^k \mathbf{A}(\boldsymbol{\xi}_i) \, \widehat{\mathbf{h}}_i(\boldsymbol{\xi}_i), \quad (35)$$

where $\{\boldsymbol{\xi}_i\}$ are the frequency variable for each direction $\{\mathbf{h}_i\}$. Eq. (35 shows that the $k$-th derivative could be read as a multi-linear operator whose *spectral kernel*

$$\mathbf{K}_k(\mathbf{u}; \boldsymbol{\xi}; \boldsymbol{\xi}_1, \dots, \boldsymbol{\xi}_k) := \widehat{\mathbf{s}_k(\mathbf{u})}(\boldsymbol{\xi} - \boldsymbol{\xi}_1 - \dots - \boldsymbol{\xi}_k) \cdot \prod_{i=1}^k \mathbf{A}(\boldsymbol{\xi}_i) \tag{36}$$

connects $(\boldsymbol{\xi}_1, \dots, \boldsymbol{\xi}_k)$ to the output function frequency $\boldsymbol{\xi}$.

Two structural facts follow immediately from Eq. (36):

1. Input function $\mathbf{u}$ decides the value $\widehat{\mathbf{s}_k(\mathbf{u})} = \mathcal{F}\big[\sigma^{(k)}\big(\mathbf{z}(\mathbf{u}, \cdot)\big)\big]$,

2. The spectral multipliers $\mathbf{A}(\boldsymbol{\xi}_i)$ depend only on the learned weights and remain diagonal in Fourier: they are independent of input $\mathbf{u}$ but does not contribute to spectral off-diagonals.

Consequently, for any $k \geq 1$:

> *Every non-diagonal spectral coupling is induced by Fourier coefficients of nonlinear activation derivatives* $\sigma^{(k)}\big(\mathbf{z}(\mathbf{u}, \cdot)\big)$ *evaluated on the current input* $\mathbf{u}$.

Higher orders $k \geq 2$ introduce convolutions of higher derivatives $\sigma^{(k)}$, but they do not mitigate the input dependence argument of spectral off-diagonals provided in Section 3.1.

**Deep FNO networks.** A full FNO is a composition of multiple FNO layers:

$$\mathcal{G}_\theta^{\text{FNO}} = \mathcal{L}_{\text{FNO}}^{(\ell)} \circ \dots \circ \mathcal{L}_{\text{FNO}}^{(1)}. \tag{37}$$

Applying the Faà di Bruno formula (Faa di Bruno, 1855) to this composition, every term in $\mathcal{D}^k \mathcal{G}_\theta^{\text{FNO}}(\mathbf{u})$ becomes a product of:

- spectral multipliers $\{\mathbf{A}_j(\boldsymbol{\xi})\}$ from the linear parts of the layers, and
- Fourier transforms of activation derivatives $\sigma^{(r)}\big(\mathbf{z}^{(j)}(\mathbf{u}, \cdot)\big)$ from intermediate layers $j$ and derivative orders $r \geq 1$.

Thus, in a *deep* FNO, all spectral off-diagonals at any order $k$ still factor through Fourier transforms of activation derivatives evaluated on intermediate pre-activations governed by the input $\mathbf{u}$. Adding layers introduces more such gates but do not mitigate the input dependence on spectral off-diagonals introduced in Section 3.1.

### D.3 THE MEANING OF INPUT DEPENDENCE OF SPECTRAL OFF-DIAGONALS WHEN LEARNING A SPECTRALLY DENSE GROUND-TRUTH OPERATORS

Let $\mathcal{T}$ be a fixed linear operator such as the position multiplier $a(x) \mapsto x^2 a(x)$ in Section 3.1. In Fourier basis, $\mathcal{T}$ is represented by a dense Toeplitz matrix such as $\mathbf{T}_n[x^2]$ as in Eq. (11), and its Fréchet derivatives are

$$\mathcal{D}\mathcal{T}(\mathbf{u}) \equiv \mathcal{T} \quad \text{for all } \mathbf{u}, \qquad \mathcal{D}^k \mathcal{T}(\mathbf{u}) \equiv 0 \quad \text{for all } k \geq 2,$$

i.e., the spectral off-diagonals of $\mathcal{T}$ are completely *independent* of the input $\mathbf{u}$.

Suppose we wish to learn $\mathcal{T}$ with robust out-of-distribution performance over a Sobolev ball

$$\mathcal{B}_B := \big\{ \mathbf{u} \in H^s : \|\mathbf{u}\|_{H^s} \leq B \big\}.$$

A natural notion of generalization is a small error in operator norm for an unseen Sobolev ball $\mathcal{B}_B$:

$$\sup_{\mathbf{u} \in \mathcal{B}_B} \frac{\big\| \mathcal{G}_\theta^{\text{FNO}}(\mathbf{u}) - \mathcal{T}\mathbf{u} \big\|_{H^s}}{\|\mathbf{u}\|_{H^s}} \leq \varepsilon. \tag{38}$$

This requires that on $\mathcal{B}_B$

- the Jacobian $\mathcal{D}\mathcal{G}_\theta^{\mathrm{FNO}}(u)$ stays close to $\mathcal{T}$, and
- all higher-order derivatives $\mathcal{D}^k \mathcal{G}_\theta^{\mathrm{FNO}}(\sqcap)$, $k \geq 2$, remain uniformly small (negligible).

Combining Eq. (36) with the composition structure above reveals a paradox:

1. If the gates $\sigma^{(k)}\big(\mathbf{z}^{(j)}(\mathbf{u}, \cdot)\big)$ and their Fourier transforms $\widehat{\mathbf{s}_k^{(j)}}(\mathbf{u})$ vary significantly with $\mathbf{u} \in \mathcal{B}_B$, then both the Jacobian and all higher-order kernels $\mathbf{K}_k(\mathbf{u}; \boldsymbol{\xi}, \boldsymbol{\xi}_1, \ldots, \boldsymbol{\xi}_k)$ vary with the input. In that case the effective spectral off–diagonals of $\mathcal{G}_\theta^{\mathrm{FNO}}$ cannot coincide with a single, input-independent Toeplitz kernel across all $\mathbf{u} \in \mathcal{B}_B$. Any dense off-diagonal pattern learned from the train distribution will be tied to the train subspace and becomes fragile under distribution shift, as observed in our experiments in Section 5.

2. If, on the other hand, we try to make these kernels effectively independent of $\mathbf{u}$ on $\mathcal{B}_B$, then Eq. (28) and Eq.(36) force all activation derivatives to be nearly constant (for $k = 1$) or nearly zero (for $k \geq 2$) on the relevant pre-activation range. In this regime, the network is forced into an almost linear operating region where

   - $\sigma'$ is approximately constant, so $\widehat{\mathbf{s}_1(u)}$ is concentrated near zero frequency and $\mathcal{D}\mathcal{L}_{\mathrm{FNO}}(\mathbf{u})$ becomes (block-)diagonal in spectral basis; and
   - $\sigma^{(k)} \approx 0$ for $k \geq 2$, so higher-order terms vanish.

   The resulting layer effectively reduces to a spectral multiplier $\mathbf{A}(\boldsymbol{\xi})$ and cannot represent a dense Toeplitz map such as $\mathbf{T}_n[x^2]$ whose off-diagonals are non-trivial and fixed independent of inputs.

In other words, for spectrally dense, position-dependent operators, FNO faces a fundamental trade-off:

- it can use strongly input-dependent gates to synthesize spectral off-diagonals, but then those off-diagonals are necessarily tied to the input distribution, or
- it can suppress the input dependence of the gates to emulate a fixed operator, but then the spectral kernel collapses towards a diagonal (or nearly diagonal) multiplier.

In neither case does a practical-size FNO realize a fixed, input-independent dense Toeplitz kernel with robust out-of-distribution generalization for spectrally dense operators, even though a sufficiently large FNO *can* parametrize the off-diagonals of the in-sample mapping on the training distribution.

**What we do (and do not) claim about FNO.** The generalized analysis in this Appendix refines the statement of Section 3.1:

- We *do not* claim that FNO cannot generate spectral off-diagonals. Eq. (12) and Eq. (36) show that at first and higher order, off-diagonals appear whenever the activation derivatives have non-trivial Fourier coefficients.
- We *do* claim that for spectrally dense operators, these off-diagonals are *always tied to input-dependent gates*. As a result, a large FNO can fit the off-diagonals of the in-sample mapping on the training subspace, but it cannot efficiently learn a fixed dense off-diagonals with robust out-of-distribution generalization on unseen function spaces which is exactly what we observe in our experiments in Section 5.

Stacking more layers introduces more input-dependent gates but does not create an input-independent spectral mixing mechanism, so depth does not mitigate this bottleneck.

## D.4 WHY KANO DOES NOT SUFFER THE SAME PROBLEM

By contrast, KANO directly learns an input-independent pseudo-differential symbol $\mathbf{p}_\theta(\mathbf{x}, \boldsymbol{\xi})$ in the dual bases via Eq. (15). KANO has a spectral kernel capable of mode mixing as it is governed by spatial basis $\mathbf{x}$ (which is convolution in spectral basis) as well as the frequency mode $\boldsymbol{\xi}$ via symbol $\mathbf{p}_\theta(\mathbf{x}, \boldsymbol{\xi})$, not only by the non-linear activation gates tied to the input function $\mathbf{u}$. Once $\mathbf{p}_\theta$ is learned, it is shared across all input functions in the ball $\mathcal{B}_B$, including even the unseen function subspace. This dual-domain, symbol-based parameterization allows KANO to learn the fixed off-diagonals of spectrally dense, position-dependent operators with robust out-of-distribution generalization, as confirmed by our experiments in Section 5.

# E   COMPUTATION AND MEMORY COMPLEXITY OF KANO

In this Appendix we compare the memory and computation complexity of KANO compared to FNO. We first quantify the per–layer costs of a single FNO layer (4) and a single KANO layer (15), and then combine them with the parameter complexity results of Section 3.2 and Section 4.2 to argue that, on the target class of variable–coefficient PDE and position–dependent dynamics, the higher per–layer cost of KANO is compensated by better model size scaling.

Throughout this appendix we work on the discrete torus $\mathbb{T}^d$ with a uniform spatial grid $\mathcal{Y} = \{\mathbf{y}_1, \ldots, \mathbf{y}_m\}$ and a truncated Fourier set $\Xi = \{\boldsymbol{\xi}_1, \ldots, \boldsymbol{\xi}_m\}$ as in Section 2.1 and Section 4.1.

## E.1   PER–LAYER FLOPS AND ACTIVATION MEMORY

We measure complexity in floating–point operations (FLOPs) and activation memory per forward pass for a single layer of each model. Backward passes in standard automatic differentiation are assumed to be within a constant factor of the forward cost and do not change the asymptotic conclusions.

**FNO layer.**   Consider a single FNO layer (4)

$$\boldsymbol{\mathcal{L}}_{\text{FNO}}(\mathbf{a})(\mathbf{x}) \;=\; \sigma\Big(\boldsymbol{\mathcal{F}}_m^{-1}\Big(\mathbf{R}_{\theta_1}(\boldsymbol{\xi}) \cdot \boldsymbol{\mathcal{F}}_m(\mathbf{a})(\boldsymbol{\xi})\Big)(\mathbf{x}) \;+\; \mathbf{W}_{\theta_2} \cdot \mathbf{a}(\mathbf{x})\Big)$$

with $C_{\text{in}}$ input channels and $C_{\text{out}}$ output channels. On a $d$-dimensional grid $\mathcal{Y}$ with $m^d$ coordinates, a single forward application of $\boldsymbol{\mathcal{L}}_{\text{FNO}}$ has the following costs:

- *FFT and inverse FFT:* $\boldsymbol{\mathcal{F}}_m$ and $\boldsymbol{\mathcal{F}}_m^{-1}$ are applied channel–wise and cost

  $$\text{FFT cost} \;\sim\; \mathcal{O}\big((C_{\text{in}} + C_{\text{out}})\, m^d \log m^d\big).$$

- *Spectral multiplier:* for each retained mode $\boldsymbol{\xi} \in \Xi$, $\mathbf{R}_\theta(\boldsymbol{\xi}) \in \mathbb{C}^{C_{\text{out}} \times C_{\text{in}}}$ is a dense matrix; multiplying by $\hat{\mathbf{u}}(\boldsymbol{\xi}) \in \mathbb{C}^{C_{\text{in}}}$ costs $\mathcal{O}(C_{\text{in}}C_{\text{out}})$ per mode, hence

  $$\text{spectral block} \;\sim\; \mathcal{O}\big(m^d\, C_{\text{in}}C_{\text{out}}\big).$$

- *Point-wise linear map:* $\mathbf{W}_\theta$ is applied at each spatial point $\mathbf{y} \in \mathcal{Y}$ as a dense matrix in channel space, costing

  $$\text{spatial linear map} \;\sim\; \mathcal{O}\big(m^d\, C_{\text{in}}C_{\text{out}}\big).$$

- *Nonlinearity:* the point-wise nonlinearity $\sigma$ is $\mathcal{O}(m^d\, C_{\text{out}})$.

Collecting terms, for fixed channel counts we obtain the per–layer forward cost:

$$\text{FLOPs}\big(\boldsymbol{\mathcal{L}}_{\text{FNO}}\big) \;\sim\; \mathcal{O}\big(m^d \log m^d + m^d\big) \;\sim\; \mathcal{O}\big(m^d \log m^d\big). \tag{39}$$

The activation memory footprint is dominated by storing $\mathbf{u}$, $\boldsymbol{\mathcal{F}}_m\mathbf{u}$, the pre–activation $\mathbf{z}(\mathbf{u}, \cdot)$ and the post–activation:

$$\text{memory}\big(\boldsymbol{\mathcal{L}}_{\text{FNO}}\big) \;=\; \mathcal{O}\big(m^d\, C_{\text{in}} + m^d\, C_{\text{out}}\big) \;\sim\; \mathcal{O}\big(m^d\big). \tag{40}$$

The parameter memory is $\mathcal{O}\big(m^d\, C_{\text{in}}C_{\text{out}}\big)$ for the spectral multipliers plus $\mathcal{O}\big(C_{\text{in}}C_{\text{out}}\big)$ for $\mathbf{W}_\theta$.

**KANO layer.**   Now consider a KANO layer (15)

$$\boldsymbol{\mathcal{L}}_{\text{KANO}}(\mathbf{a})(\mathbf{x}) \;=\; \boldsymbol{\Phi}_{\theta_1}\Big(\boldsymbol{\mathcal{F}}_m^{-1}\big[\, \boldsymbol{p}_{\theta_2}(\mathbf{x}, \boldsymbol{\xi}) \,*\, \boldsymbol{\mathcal{F}}_m(\mathbf{a})(\boldsymbol{\xi})\,\big](\mathbf{x})\,,\; \mathbf{a}(\mathbf{x})\Big),$$

where $\boldsymbol{p}_{\theta_2}(\mathbf{x}, \boldsymbol{\xi})$ is implemented by a width–fixed KAN symbol network and $\boldsymbol{\Phi}_{\theta_1}$ is another width–fixed KAN activation network. Using Kohn–Nirenberg quantization (16):

$$\boldsymbol{\mathcal{F}}_m^{-1}\big[\, \boldsymbol{p}(\mathbf{x}, \boldsymbol{\xi}) \,*\, \boldsymbol{\mathcal{F}}_m(\mathbf{a})(\boldsymbol{\xi})\,\big](\mathbf{x}) := \left(\frac{h}{L}\right)^d \sum_{\boldsymbol{\xi} \in \Xi} \sum_{\mathbf{y} \in \mathcal{Y}} e^{i(\mathbf{x}-\mathbf{y})\cdot\boldsymbol{\xi}}\, \boldsymbol{p}(\mathbf{x}, \boldsymbol{\xi})\, \mathbf{a}(\mathbf{y}),$$

on a $d$-dimensional grid $\mathcal{Y}$ with $m^d$ points and a retained frequency set $\Xi$ with $m^d$ modes, a single forward application of $\boldsymbol{\mathcal{L}}_{\text{KANO}}$ has the following costs:

- *KN quantization (double sum):* for each output location $\mathbf{x}$ we evaluate the double sum over $\boldsymbol{\xi} \in \Xi$ and $\mathbf{y} \in \mathcal{Y}$, i.e. $m^{2d}$ terms per $\mathbf{x}$. Each term involves a dense matrix–vector product $\boldsymbol{p}(\mathbf{x}, \boldsymbol{\xi})\, \mathbf{a}(\mathbf{y})$ of cost $\mathcal{O}(C_{\text{in}} C_{\text{out}})$. Summed over all $\mathbf{x} \in \mathcal{Y}$ this yields

$$\text{KN operator} \ \sim \ \mathcal{O}\big(m^{3d}\, C_{\text{in}} C_{\text{out}}\big).$$

- *Symbol network evaluation:* the symbol KAN $\boldsymbol{p}_{\theta_2}$ is evaluated once per pair $(\mathbf{x}, \boldsymbol{\xi}) \in \mathcal{Y} \times \Xi$, i.e. at $m^{2d}$ points. Let $P_p$ denote the cost of a single forward evaluation of $\boldsymbol{p}_{\theta_2}$. Then

$$\text{symbol KAN} \ \sim \ \mathcal{O}\big(m^{2d}\, P_p\big).$$

- *Activation KAN:* the activation network $\boldsymbol{\Phi}_{\theta_1}$ is applied point-wise at each spatial point $\mathbf{x} \in \mathcal{Y}$. Let $P_\Phi$ denote the cost of one forward pass of $\boldsymbol{\Phi}_{\theta_1}$. Then

$$\text{activation KAN} \ \sim \ \mathcal{O}\big(m^d\, P_\Phi\big).$$

Collecting terms, for fixed channel counts and fixed KAN architectures we obtain the per–layer forward cost:

$$\text{FLOPs}\big(\mathcal{L}_{\text{KANO}}\big) \ \sim \ \mathcal{O}\big(m^{3d}\, C_{\text{in}} C_{\text{out}} + m^{2d} P_p + m P_\Phi\big). \tag{41}$$

The activation memory footprint is dominated by storing the feature maps and, if materialized, the symbol grid $\boldsymbol{p}_{\theta_2}(\mathbf{x}, \boldsymbol{\xi})$:

$$\text{memory}\big(\mathcal{L}_{\text{KANO}}\big) \ \sim \ \mathcal{O}\big(m^d\, C_{\text{in}} + m^d\, C_{\text{out}} + m^{2d}\, C_{\text{in}} C_{\text{out}}\big) \ \sim \ \mathcal{O}\big(m^{2d}\, C_{\text{in}} C_{\text{out}}\big), \tag{42}$$

in addition to the parameter size of the KAN subnetworks.

**Inference cost of compact KAN subnetworks.** For completeness, we quantify the inference cost $P_p$ and $P_\Phi$ of the KAN subnetworks. Consider a fully connected KAN layer with input width $d_{\text{in}}$, output width $d_{\text{out}}$, and $G$ basis functions per edge. Each edge $(j \to i)$ carries a learnable univariate function $f_{ij} : \mathbb{R} \to \mathbb{R}$ represented as

$$f_{ij}(x) \ = \ \sum_{g=1}^{G} w_{ijg}\, \phi_g(x),$$

where $\{\phi_g\}_{g=1}^{G}$ are fixed basis functions (e.g. B–splines or rational functions) and $w_{ijg}$ are learned coefficients. Evaluating $f_{ij}(x_j)$ for a given scalar input $x_j$ requires computing the active basis functions $\phi_g(x_j)$ and a dot product over $G$ elements.

A single KAN layer thus computes, for each output coordinate $i$,

$$y_i \ = \ \sum_{j=1}^{d_{\text{in}}} f_{ij}(x_j),$$

and the total cost of one forward pass through this layer would be $\mathcal{O}\big(d_{\text{out}}\, d_{\text{in}}\, G\big)$, up to lower order terms from basis evaluation. The parameter count of this layer is of the same order, $\mathcal{O}(d_{\text{out}}\, d_{\text{in}}\, G)$. Under the assumption of compact KAN subnetworks, widths, depths, and the number of basis functions $G$ are all small and independent to the operator resolution $m$. If we denote by $N$ an upper bound on their layer widths and by $L_{\text{KAN}}$ their depth, then their total inference costs satisfy

$$P_p, \ P_\Phi \ = \ \mathcal{O}\big(L_{\text{KAN}}\, N^2\, G\big).$$

In the per–layer KANO complexity (41) the KAN subnetworks contribute only a constant–factor overhead that does not grow with the spatial or spectral resolution, hence for large resolution $m \gg N, L_{\text{KAN}}$ the dominant term would be the KN quantization term $\mathcal{O}(m^{3d} C_{\text{in}} C_{\text{out}})$.

### E.2 COMPUTATION COMPLEXITY AS A FUNCTION OF ACCURACY FOR POSITION-DEPENDENT DYNAMICS

As apparent in Eq. (39 and Eq. (41), KANO requires orders heavier FLOPs compared to FNO due to the expensive double sum nature of its KN quantization, when two models are of similar size. For small resolution $m$, the inference cost $P_p$ and $P_\Phi$ still set hard lower bound on the computation

complexity as well. However, one of the main arguments we make in this work is that KANO enjoys incomparable parameter efficiency compared to FNO on the target class of spectrally dense operators such as position-dependent dynamics: parameter complexity of KANO scales polynomially where that of FNO can scale super-exponentially on a spectrally dense ground-truth operator. Therefore, KANO's expensive FLOPs requirement can be compensated by parameter efficiency where FNO suffers the curse of dimensionality.

For a position-dependent dynamics $\mathcal{G}$ and accuracy $\varepsilon > 0$, we write $|\Theta_{\text{FNO}}(\varepsilon)|$ and $|\Theta_{\text{KANO}}(\varepsilon)|$ for the smallest parameter counts required by FNO and KANO respectively to achieve $\|\mathcal{G} - \mathcal{G}_\theta\| \leq \varepsilon$, following Theorem 1 and Theorem 2. Also, let $T_{\text{FNO}}(\varepsilon)$ and $T_{\text{KANO}}(\varepsilon)$ denote the cost of one forward (or forward+backward) pass of an FNO or KANO architecture chosen to achieve accuracy $\varepsilon$ on the ground-truth operator. We do not attempt to model the number of optimization steps; instead, we focus on how the cost of a *single* training or inference step scales with $\varepsilon$. Assuming a model must perform at least a constant number of FLOPs per parameter in each step, lower bound for computation complexity can be estimated as:

$$T_{\text{FNO}}(\varepsilon) \gtrsim c_0 |\Theta_{\text{FNO}}(\varepsilon)|, \qquad T_{\text{KANO}}(\varepsilon) \gtrsim c_0' |\Theta_{\text{KANO}}(\varepsilon)| \tag{43}$$

for constants $c_0, c_0' > 0$. Such lower bound assumption is not universal for any neural network, but in FNO architecture all its parameters equally appear in the matrix computation via the dense parameter block $\mathbf{R}$ and $\mathbf{W}$; especially for implementations that evaluate and update all parameters in every step. Hence, computation complexity lower bound assumption by its model size is reasonable for FNO.

**Computation complexity of FNO by accuracy for position operators from Theorem 1.** For the position–multiplier $\mathcal{M}(\mathbf{x}) := x_1^{\alpha_1} \cdots x_d^{\alpha_d}$ of total degree $M := |\alpha| \geq 1$, the restatement of Theorem 1 in Appendix C.3 shows that an FNO $\mathcal{G}_\theta^{\text{FNO}}$ that achieves

$$\|\mathcal{M} - \mathcal{G}_\theta^{\text{FNO}}\|_{H^s} \leq \varepsilon$$

on band–limited inputs can suffer curse of dimensionality:

$$|\Theta_{\text{FNO}}(\varepsilon)| \sim \exp(c\,\varepsilon^{-Md/s}), \tag{44}$$

for some constant $c > 0$ depending only on geometric constants of Sobolev space $H^s$. From Eq. (43), the computation complexity of FNO by accuracy on $\mathcal{M}$ can be written as:

$$T_{\text{FNO}}(\varepsilon) \gtrsim c_0 |\Theta_{\text{FNO}}(\varepsilon)| \sim \mathcal{O}\big(\exp(\varepsilon^{-Md/s})\big) \tag{45}$$

**Computation complexity of KANO by accuracy for position operators from Theorem 2.** On the other hand, Theorem 2 and Corollary 1 show that as the KANO projection of $\mathcal{M}$ yields symbols $\mathbf{p}_{\mathcal{M}}(\mathbf{x}, \boldsymbol{\xi})$ of sufficient smoothness, KANO can achieve $\|\mathcal{M} - \mathcal{G}_\theta^{\text{KANO}}\| \leq \varepsilon$ with

$$|\Theta_{\text{KANO}}(\varepsilon)| \sim \mathcal{O}\big(\varepsilon^{-\beta}\big), \tag{46}$$

where $\beta = d/(2s_p)$ or $\beta = d/(2r)$ is a geometric exponent determined by the symbol regularity (Wang et al., 2024). From Eq. (43), the computation complexity of KANO by accuracy on $\mathcal{M}$ can be written as:

$$T_{\text{KANO}}(\varepsilon) \gtrsim c_0' |\Theta_{\text{KANO}}(\varepsilon)| \sim \mathcal{O}\big(\exp(\varepsilon^{-\beta})\big) \tag{47}$$

Therefore, when the ground-truth operator is spectrally dense yet its KANO projection give smooth enough symbol so that both Theorem 1 and Theorem 2 are effective (for instance, position-dependent dynamics), lower bound of computation complexity of FNO can prevail over that of KANO because of the curse of dimensionality discussed in Section 3.2.

We emphasize that this is an asymptotic statement under the specific operator class where both Theorem 1 and Theorem 2 hold. For operators that are nearly spectrally diagonal (e.g. standard FNO benchmarks), FNO is theoretically much faster than KANO when the two models are in similar size. However, when learning a spectrally dense, variable–coefficient operators to high precision, KANO's parameter efficiency can, in theory, compensate for its expensive double sum KN quantization while FNO faces a super–exponential growth in model size hence per–step FLOPs to achieve the same level of accuracy.

# F  USE OF LARGE LANGUAGE MODEL

Large Language Model (LLM) is used to polish the writing in this paper, such as detecting grammar errors and typos. LLM is also used to search for appropriate references for correct citations, and all the proposed references are fully inspected and verified before citing in the paper.

