# OpenReview forum: "KANO: Kolmogorov-Arnold Neural Operator"
_ICLR.cc/2026/Conference — ICLR 2026 Poster_

### Official Review · Reviewer_mZHA · 2025-10-29

**Soundness:** 3
**Presentation:** 4
**Contribution:** 3
**Rating:** 6
**Confidence:** 4

**Summary:**

The paper introduces the Kolmogorov–Arnold Neural Operator (KANO), a novel neural operator that learns pseudo-differential symbols in both spatial and spectral domains using Kolmogorov–Arnold Networks (KANs). The authors identify a theoretical limitation in Fourier Neural Operators (FNOs), the "pure-spectral bottleneck", which restricts their expressivity for position-dependent dynamics. They provide a rigorous theoretical analysis showing that FNOs suffer from super-exponential scaling in model size for such operators, while KANO achieves polynomial scaling. Empirical validation includes synthetic PDE operator learning and quantum Hamiltonian learning, where KANO demonstrates strong out-of-distribution generalization, high parameter efficiency, and symbolic recovery of operator forms.

**Strengths:**

This paper makes a significant and well-structured contribution to SciML. It identifies a fundamental flaw in a popular architecture, proposes a principled solution, and provides convincing evidence of its superiority. The authors present a rigorous theoretical diagnosis of FNOs, exposing the “pure-spectral bottleneck.” They show that for position-dependent dynamics, including quantum mechanics and fluid flow, FNOs suffer from super-exponential scaling. This is framed not as a minor drawback but as a deep architectural flaw.

In response, the KANO is both novel and well aligned with the mathematics of the problem. By jointly parameterizing operators in spatial and spectral domains through a pseudo-differential framework, KANO represents each term in its naturally sparse basis. This dual-domain approach provides the right inductive bias for robust learning.

The empirical results reinforce the theory with striking effect. KANO does not merely improve modestly over FNO but shows dramatic gains: flawless out-of-distribution generalization, 0.03% of the parameters, and symbolic operator recovery accurate to the fourth decimal. This interpretability moves the model from black-box approximation toward genuine scientific discovery.

**Weaknesses:**

The main weakness is the narrow experimental validation. Results are confined to 1D synthetic operators and quantum systems, leaving scalability to high-dimensional PDEs untested. Absent are standard benchmarks such as 2D Navier–Stokes or 3D elasticity, along with any runtime or stability analysis, making the practical utility unclear.

Comparisons are also limited. Most results benchmark only against vanilla FNO, whose flaws are already established, creating a straw-man dynamic. Stronger baselines like U-FNO, AM-FNO, PDNO, or multi-scale FNO variants are missing, making it hard to judge whether KANO’s advantage stems from its design or from outdated comparisons.

Finally, while symbolic recovery is compelling, it remains demonstrated only in simple, smooth cases. Whether this interpretability extends to non-smooth, higher-dimensional, or more complex operators is left open. Overall, the idea is strong, but the evidence is not yet broad enough to confirm its readiness for widespread use.

**Questions:**

How does KANO perform on widely used PDE benchmarks such as 2D Navier–Stokes or heterogeneous material modeling?

Could you include comparisons with stronger baselines like U-FNO, PDNO, or adaptive FNO variants?

What are the runtime and memory costs of KANO compared to FNO during training and inference? Given the dual-domain computations, how does training time scale?

How robust is symbolic recovery when operator coefficients are non-smooth or discontinuous?

What optimization challenges or instabilities were encountered when training the KAN sub-networks within the operator framework, and how were they addressed?

Did you perform ablation studies to understand the individual contributions of the dual-domain design versus the KAN sub-networks? For instance, what is the performance of a KANO variant that uses MLPs instead of KANs?

---

> ### Author Response · Authors · 2025-11-21
>
> We deeply appreciate your review and faithfully answer to all your questions.
> ## **On the absence of 2D/3D benchmarks**
> Thanks for pointing out this issue. 2D/3D problems were not considered when we designed the current KANO, as our primary goal was to demonstrate the core math concept of symbolic interpretability and dual-domain expressivity. As a result, current KANO cannot effectively handle 2D/3D because it excludes lift/projection networks, which are central for high-dim performance. We added Remark 3 in Section 4.2 to address this.
>
> That said, extending KANO to 2D/3D PDE tasks is highly doable, since the core mathematics of dual-domain expressivity is agnostic to dimensionality.  We are very interested in this exciting future direction and will happily share more results in the near future.
>
> ## **Strengthening baselines with stronger FNO variants**
>   Thank you for your suggestion, we have added the three additional baselines you have suggested in our benchmark of Section 5.1 and we brief the result:
> | Model | 𝒢₁ B/A | 𝒢₂ B/A | 𝒢₃ B/A |
> | --- | --- | --- | --- |
> | FNO (566k) | 15.53 | 8.21 | 7.14 |
> | U-FNO (579k) | 8.21 | 41.65 | 3.16 |
> | AM-FNO (548k) | 19.35 | 13.75 | 25.69 |
> | PDNO (538k) | 4.5 | 6.3 | 6.7 |
> | KANO (152) | 1.38 | 1.19 | 1.03 |
> Here B/A is the ratio of average loss between train and test dataset to quantify the generalization.
>
> U-FNO and AM-FNO showed no better out-of-distribution (OOD) fragility than FNO. PDNO of pseudo-differential operator framework as KANO showed the best result but not as robust as KANO. This confirms that the dual-domain structure is indeed judicious for robust generalization on position-dependent operators, while localized enhancements are not enough to resolve the pure-spectral bottleneck of FNO.
>
> ## **On the complexity analysis of KANO**
>   Thank you for pointing out the missing complexity analysis. We have now added a detailed analysis in Appendix E.
>   In short:
> - KANO is more expensive than FNO **when they have the same model size**, primarily due to the double sum block in Eq. (16). Unfortunately, we could not empirically test our analyses since public pykan library is not optimized on GPU. Actually, optimizing KAN on GPU is still a very active research direction [1-3], and we expect the computing efficiency gap between KANO and KAN to be narrowed down with the progress in near future.
> - In practice, KANO needs order-of-magnitude fewer parameters than FNO for the target operator class in this paper. **As a result, to solve the same problem, KANO is often be much more computing- and memory-efficient due to this superior parameter efficiency**.
>
> [1] https://github.com/Blealtan/efficient-kan
>
> [2] "Kolmogorov-arnold networks are radial basis function networks."
>
> [3] "Chebyshev polynomial-based kolmogorov-arnold networks: An efficient architecture for nonlinear function approximation."
>
> ## **On the robustness of symbolic recovery**
>   Thank you for this insightful question. Empirically, recent benchmark[4] shows that symbolic recovery of KAN degrades for localized irregularities or highly oscillatory targets. However, a number of recent works improve KAN’s symbolic capacity and each can be seamlessly integrated into KANO. We have addressed this matter at the end of Section 4.1, citing all the notable recent improvements.
>
> [4] "Kan versus mlp on irregular or noisy functions."
>
> ## **On the optimization challenges when training KANO**
>   Thank you for this thoughtful question. In our runs, pykan library introduced noticeable overhead due to limited parallelization and expensive internal calls. We insisted on using pykan because of its symbolic‑recovery tooling, but using efficient variants of KAN would alleviate this problem. Also, consistent with recent analyses [5], training depended strongly on learning rate (LR) schedules and on the shape of the KANs. We found that warming up with a few epochs of large LR and then dropping an order to further train for longer epochs greatly improved training.
>
> We also would like to propose a precursor technique of first training KANO with MLP subnetworks and then swap with KAN after distilling, if anyone should encounter challenges in training KANO.
>
> [5] "On training of kolmogorov-arnold networks."
> ## **On the ablation studies replacing KAN with MLP**
>   We greatly appreciate your inspiring suggestion. We trained MLP variant of KANO and tested on our benchmarks. In brief, KANO-MLP demonstrates robust OOD behavior comparable to KANO, and achieves the same order state infidelity.
> | Model | 𝒢₁ B/A | 𝒢₂ B/A | 𝒢₃ B/A | DW | NLSE |
> | --- | --- | --- | --- | --- | --- |
> | KANO (152) | 1.38 | 1.19 | 1.03 | 6.3$\times 10^{-6}$ | 5.6$\times 10^{-6}$ |
> | KANO_mlp (2k) | 1.96 | 1.80 | 1.91 | 7.7$\times 10^{-6}$ | 8.5$\times 10^{-6}$ |
>
> This suggests that KANO’s robust OOD behavior mainly stems from the dual‑domain architecture. We included this ablation study and results in Section 5, and deeply appreciate your contribution.

---

### Official Review · Reviewer_bFzm · 2025-11-01

**Soundness:** 3
**Presentation:** 4
**Contribution:** 3
**Rating:** 4
**Confidence:** 4

**Summary:**

The authors proposed the Kolmogorov-Arnold Neural Operator KANO that uses the KAN to construct the neural operator, which has very solid advantages over FNO both from theory and experiment.

**Strengths:**

1. The explanation of why the FNO have problems is very clear. Strongly support the contribution and advantages of the proposed KANO.

**Weaknesses:**

1. The idea is quite natural, if picky. But it does not affect the contribution.
2. The experiment is quite limited. I think this method should benefit a lot from the flexibility of the KAN. Extra experiments of diverse operators can make this paper attractive to a broader audience.

**Questions:**

1. Are there other results that demonstrate the advantages of this method?

2. For the demonstration of the spectrum limitation of FNO, the author uses the linear approximation. Will the off-diagonal contribution arise in higher orders? What will that affect in the argument in this paper?

---

> ### Author Response · Authors · 2025-11-21
>
> We deeply appreciate your review and faithfully answer to all your questions.
>
> ## Additional results that demonstrate the advantage of KANO’s dual-domain architecture
>
>    Thank you for your request on additional results to further demonstrate the advantage of our work. We have two more results to support the benefit of KANO’s dual-domain architecture: (1) **strengthened baselines** with stronger FNO variants, (2) an **ablation study** that disentangles KAN’s capacity from KANO.
>
>    **(1)** In the revised manuscript of Section 5.1 we have added baselines of U‑FNO, AM-FNO and PDNO
>
> | Model (params) | 𝒢₁ B/A | 𝒢₂ B/A | 𝒢₃ B/A |
> | --- | --- | --- | --- |
> | FNO (566k) | 15.53 | 8.21 | 7.14 |
> | U-FNO (579k) | 8.21 | 41.65 | 3.16 |
> | AM-FNO (548k) | 19.35 | 13.75 | 25.69 |
> | PDNO (538k) | 4.5 | 6.3 | 6.7 |
> | KANO (152) | 1.38 | 1.19 | 1.03 |
>
> Here B/A is the ratio of average loss between Group B out-of-distribution (OOD) dataset and Group A train distribution dataset, to quantify the generalization of each model.
>
>   Opposed to our expectation, U-FNO and AM-FNO showed no better OOD fragility than FNO, while PDNO, which is also based on the pseudo-differential operator framework as KANO, showed the best result in the FNO-family, yet not as robust as KANO. This confirms that the **dual-domain structure of pseudo-differential operator is indeed judicious for robust generalization on position-dependent operators**, while primarily relying on only spectral basis makes the model fragile out of train distribution even with localized enhancements.
>
>   **(2)** Also, to understand to what extent KANO benefits from the capacity of KAN, we performed ablation studies by replacing KAN subnetworks with MLPs (KANO-MLP) in Section 5 of the revised manuscript.
>
> | Model | 𝒢₁ B/A | 𝒢₂ B/A | 𝒢₃ B/A | DW | NLSE |
> | --- | --- | --- | --- | --- | --- |
> | KANO (152) | 1.38 | 1.19 | 1.03 | 6.3$\times 10^{-6}$ | 5.6$\times 10^{-6}$ |
> | KANO_mlp (2k) | 1.96 | 1.80 | 1.91 | 7.7$\times 10^{-6}$ | 8.5$\times 10^{-6}$ |
>
> In brief, KANO-MLP demonstrates robust OOD behavior comparable to KANO in our Section 5.1 benchmark, and achieves the same order state infidelity in our Section 5.2 benchmark. This suggests that **KANO’s robust OOD behavior mainly stems from the dual‑domain architecture**. We included this ablation study and results in Section 5, and deeply appreciate your contribution.
>
> ## On the higher order contributions to spectral off-diagonals in our arguments of Section 3.1
>
>   We thank you for this thoughtful question, which helped us clarify and strengthen the logic in Section 3.1. In brief, our conclusion is that **higher‑order contributions to spectral off-diagonals do not resolve the structural limitation of a single layer FNO, and also for the multi-layer FNO**.
>
>   In the revised manuscript we have added a dedicated section, Appendix D: Additional Details on the Pure‑Spectral Bottleneck of FNO. Key results are:
>
> - **Higher order Frechet derivatives of a single FNO layer. (Proposition 1)**
> - **Persisting distribution-shift fragility on its spectral off-diagonals.**
> - **Expansion to multi-layer FNO**
>
> In the Appendix we further explain why the structure leads to brittle OOD performance on spectrally dense operators, and why KANO is free of such problem due to its input‑independent spectral mixing mechanism. We are very grateful for your contribution on this, which significantly improved the completeness and clarity of our theoretical discussion.

---

### Official Review · Reviewer_Nuuh · 2025-11-01

**Soundness:** 3
**Presentation:** 3
**Contribution:** 3
**Rating:** 8
**Confidence:** 4

**Summary:**

The authors consider the neural operator learning setup and consider a class of operators which involves spatial multipication and term it as position dependent dynamics to highlight the shortcoming of the existing fourier neural operator framework. Since the multiplication and differentiation have a dual relationship under the fourier transform, the authors argue that the considered class defined by the multiplication map is just a two step up-shift sparse matrix in the spatial basis, however, takes the form of a dense Teoplitz matrix in the spectral basis and the fourier operator cannot efficiently approaimte the off-diagonal elements of this matrix. The authors first show that this position dependent dynamics will induce a super-exponential scaling in the fourier operator. To adress this issue the authors propose the KANO framework which replaces the nodes of standard MLP with simple sum operations and learn the univariate 1D functions. This reparameterization is similar in expressivity to MLPs upto constant depth and width empirically. Also, they propose using a KAN sub-network jointly parameterized by both spatial and spectral basis and enjoys sparse representations in both spatial and spectral domains. They also discuss that the projection error scales efficieitnyl by its width and the latent network error also scales efficiently independent of the projection error and thus, it is more efficient as compared to the FNO for the considered class. Furthermore, they also verify this empirically by considering a dataset of the considered class where they observe that KANO outperforms FNO with much lesser size required.

**Strengths:**

Investigating the bottlenecks of the FNO by considering this position dependent dynamics class and then drawing the inference that this will incur dense representation in one basis and sparse in another is really interesting. Further utilising the KA network based parameterization and resolving this issue is also interesting and provides new insights for various scenarios of this operator learning framework. Further verification by empirically investigating this under different testing /training families and demonstrating that KANO can outperform FNO with much lesser paramter. Also the section 5.2 further including two-position dependent quantum dynamics benchmark further strenghten their claims.

**Weaknesses:**

No major weaknesses but the authors could have also compared the FNO and proposed KANO on the original equations considered in the FNO paper to see how does it perform in that setup to understand its overall use case as against this specific class. It is a bit unclear how this new framework will do on the  other tasks that FNO can perform very efficiently.

**Questions:**

Since the FNO based layers have been also used for a lot of other tasks like weather forecasting [1] or like token mixers for transformers [2], do the authors have any comments on the broader applicability of this framework?

[1] Fourcastnet: A global data-driven high-resolution weather model using adaptive fourier neural operators.
[2] Adaptive Fourier Neural Operators: Efficient Token Mixers for Transformers. ICLR 2022.

---

> ### Author Response · Authors · 2025-11-21
>
> We deeply appreciate your review and faithfully answer to your question.
>
> ## **On the absence of FNO benchmarks**
>
>   We would first like to respectfully clarify that the reason why FNO benchmarks are not included in this work is because the **efficient regimes of FNO and KANO are very different, complementary even** Below are the results of the strengthened baselines on our benchmarks:
>
> | Model (params) | 𝒢₁  Train | 𝒢₁ Test | 𝒢₂ Train | 𝒢₂ Test | 𝒢₃ Train | 𝒢₃ Test |
> | --- | --- | --- | --- | --- | --- | --- |
> | FNO (566k) | 6.36 | 98.8 | 10.6 | 87.0 | 11.4 | 81.4 |
> | U-FNO (579k) | 2.79 | 22.9 | 8.14 | 339 | 92.4 | 292 |
> | AM-FNO (548k) | 1.08 | 20.9 | 1.20 | 16.5 | 1.16 | 29.8 |
> | PDNO (538k) | 1.41 | 6.31 | 1.92 | 12.1 | 4.03 | 27.2 |
> | KANO (152) | 1.04 | 1.44 | 0.629 | 0.749 | 0.716 | 0.737 |
>
> KANO clearly prevails against stronger FNO baselines with localized enhancements, especially on the out-of-distribution robustness, with incomparably smaller model size.
>
> On the other hand, on its own benchmark, FNO is not only empirically demonstrated, but also has been mathematically proven to be near-optimal [1]: FNO prevails on the spectrally sparse operators (FNO benchmarks), and KANO takes over where the operators are spectrally dense variable-coefficient PDEs. **The core of our work, along with the symbolic interpretability, is to both theoretically and empirically validate KANO’s efficient regime, hence we did not consider the well-established efficient ground of FNO, which is complementary to our target use-case**.
>
> [1] Kovachki, Nikola, Samuel Lanthaler, and Siddhartha Mishra. "On universal approximation and error bounds for Fourier neural operators." *Journal of Machine Learning Research* 22.290 (2021): 1-76.
>
> ## **On the broad applicability of KANO**
>
>   We greatly appreciate your thoughtful inquiry on the potential use-case of KANO. We envision KANO to be beneficial wherever the governing operators are **variable‑coefficient PDEs**. Examples include:
>
> - **Quantum experiments and control**: reconstructing unknown Hamiltonians and predicting long term state evolution from the realistic projective measurements. In contemporary quantum apparatus the target state is often confined to be one-dimensional (such as photonic/atomic qubits), which also suits our benchmark.
> - **Optics/photonic devices**: wave propagation with spatially varying refractive index profiles, where mode mixing is governed by the local structure and the photon states are often confined to be one-dimensional.
> - **System identification from sparse experiments**: when an interpretable operator network that robustly extrapolates to the unseen instances is required for design or control.
>
> Across these settings, we expect KANO’s dual‑domain architecture with its symbolic interpretability will offer good applicability.

---

> > ### Comment · Reviewer_Nuuh · 2025-11-25
> >
> > Thanks for the rebuttal. I would like to maintain my initial rating.

---

> > > ### Author Response · Authors · 2025-11-26
> > >
> > > Thank you so much for your decision, we greatly appreciate it.

---

### Author Response · Authors · 2025-11-21
**Summary of Rebuttal**

We gratefully appreciate all the insightful reviews on our paper. We saw great opportunities to further improve and enhance our work thanks to the reviewers’ efforts, and below is the list of reflections in the revised manuscript:

- **Section 1** Introduction is revised to clarify the scope and intended use-case of KANO.
- **Remark 1** at the end of **Section 3.2** added for the completeness of the upper bound analysis of the section.
- **Remark 3** at the end of **Section 4.2** added to clarify the scope of KANO
- **Appendix D** is added to expand the argument in Section 3.1 into arbitrary high order expansion and multi-layer FNO, to show our logics stay intact in the generalized setting.
- **Appendix E** is added to provide theoretical analysis on the computation and memory complexity of KANO compared to FNO (+Remark 2)
- Latter part of **Section 4.1** is revised to comment on the robustness of symbolic recovery.
- Additional baselines of FNO variants in **Section 5.1**.
- Additional ablation study in **Section 5** by a KANO variant with MLP subnetworks to understand individual contributions of the dual-domain architecture versus KAN subnetworks.

Changes in the main manuscript are highlighted in blue color.

---

### Meta-Review · Area_Chair_L1Ss · 2026-01-06

**Summary:**

This paper proposes the Kolmogorov–Arnold Neural Operator (KANO), which is a neural operator that learns pseudo-differential symbols in both spatial and spectral domains using Kolmogorov–Arnold Networks (KANs). The main concern raised by other reviewers was the limited experiments. The authors added baselines and also conducted an ablation study. These are considered to address the reviewers' concerns to a certain extent, and the scores may be raised. Considering this, I recommend acceptance of this paper.

**Reviewer Concerns:**

One reviewer pointed out that the idea is natural, but also noted that this is not a drawback. The main concern was that the experiments were limited. To address this concern, the authors strengthened baselines with FNO variants and conducted an ablation study. While the additional experiments may not be thorough, they were a reasonable amount for the limited period.

**Reviewer Scores:**

The original scores were 8-6-4. Discussions with reviewers were partially conducted, and the reviewer who gave a score of 8 maintains this rating. The main concern was the limited scope of the experiments, and to address this concern, the authors added some experiments. The additional experiments considered to address the reviewers' concerns at least to some extent, and the score may have been revised to 8-6-6 or a similar level.

---

### Decision · Program_Chairs · 2026-01-26

Accept (Poster)